# Membrane insertion of mitochondrial-encoded proteins regulates ribosome decoding speed

Thomas Schöndorf[1,2,7], Valentyn Petrychenko [3,7], Ilgin Kotan [4], Drishan Dahal [1], Natalia Napieraj [1], Luis Daniel Cruz-Zaragoza [1], Cong Wang[1], Oliver Urbach[1], Tanja Gall[1], Sven Dennerlein[1], Günter Kramer [4], Niels Fischer [3] ✉ & Peter Rehling [1,2,5,6] ✉

The human mitochondrial genome encodes 13 subunits of the oxidative phosphorylation system essential for energy metabolism to drive cellular activities. Translation of 11 mRNAs by membrane-bound ribosomes is coupled to insertion of the nascent polypeptides into the inner membrane aided by the OXA1L insertase. To this end, the mechanism of membrane insertion of nascent polypeptides and the functional link to the translation process are not sufficiently understood. Here, we applied ribosome profiling to assess translation dynamics in combination with cryo-electron microscopy analysis of a COX1 ribosome–nascent chain complex to visualize cotranslational folding of the nascent chain. We find that the membrane topology of the translation product impacts translation speed and that positioning of amphipathic helices in the ribosome vestibule induces structural changes, correlating with translation pausing events. Thus, our findings reveal a link between translation process and folding and membrane insertion of nascent polypeptides at the inner mitochondrial membrane.

The mitochondrial genome encodes 13 core subunits of the oxidative phosphorylation (OXPHOS) system required for cellular energy homeostasis. Accordingly, mitochondrial translation is essential for building the OXPHOS machinery[1–3]. Translation occurs on inner mitochondrial membrane (IMM)-bound ribosomes. The mitochondrial-encoded polypeptides are cotranslationally inserted into the lipid phase of the inner membrane by the conserved OXA1L insertase[2–5]. In the membrane, the newly synthesized polypeptides associate with specific membrane proteins that stabilize the nascent chain and facilitate early steps of the biogenesis process together with nuclear-encoded, imported subunits. Yet, the spatiotemporal coordination of translation with membrane protein biogenesis remains enigmatic[4,6,7].

Recent cryo-electron microscopy (cyo-EM) studies of human mitochondrial ribosome–nascent chain complexes (mtRNCs) bound to OXA1L revealed OXA1L orientation and shape and an mL45-mediated nascent chain gating mechanism[4,8,9]. However, these structures derive

[1]Department of Cellular Biochemistry, University Medical Center Goettingen, Goettingen, Germany. [2]Cluster of Excellence 'Multiscale Bioimaging: from Molecular Machines to Networks of Excitable Cells' (MBExC), University of Goettingen, Goettingen, Germany. [3]Project Group Molecular Machines in Motion, Department of Physical Biochemistry, Max Planck Institute for Multidisciplinary Sciences, Goettingen, Germany. [4]Center for Molecular Biology of Heidelberg University (ZMBH) and German Cancer Research Center (DKFZ), DKFZ-ZMBH Alliance, Heidelberg, Germany. [5]Fraunhofer Institute for Translational Medicine and Pharmacology ITMP, Translational Neuroinflammation and Automated Microscopy, Goettingen, Germany. [6]Max Planck Institute for Multidisciplinary Sciences, Goettingen, Germany. [7]These authors contributed equally: Thomas Schöndorf, Valentyn Petrychenko. ✉e-mail: niels.fischer@mpinat.mpg.de; peter.rehling@medizin.uni-goettingen.de

from heterogeneous mixtures of all 13 mitochondrial translation products, limiting detailed analysis of cotranslational insertion, including nascent chain folding sites and coupling to translation[4,8,9].

Studies on bacterial ribosomes show that transmembrane helices can form in the lower tunnel, shaping nascent protein folding[10]. In contrast, mammalian mtRNCs contain two mL45-mediated constrictions that restrict lower-tunnel folding, while it is unclear whether folding can occur in the uL23m/uL24m/mL45 vestibule[4]. This mitochondrial-specific vestibule is more extended than in bacteria, creating a gap to OXA1L that may facilitate translation-state-dependent factor access but also increases the risk of off-pathway interactions and aggregation. Conversely, nascent chains may affect translation. Furthermore, cotranslational insertion of OXPHOS subunits involves specific assembly factors[11–13], whose structural roles remain unresolved; for example, nascent COX1 associates with OXA1L, C12ORF62 (COX14) and MITRAC12 (COA3) to form the MITRAC complex, enabling translational plasticity[2,14].

In addition to the abovementioned structural differences between mitochondrial and bacterial ribosomes, mitochondrial translation appears to operate without polysomes and relies on a single tRNA$^{Met}$ for both initiation and elongation, with leaderless mRNAs that lack canonical Shine–Dalgarno sequences[15]. Moreover, the cotranslational insertion of the multimembrane-spanning OXPHOS subunits faces another level of complexity as it needs to be coordinated with the availability of imported subunits and cofactor biosynthesis, highlighting the intricate coordination between nuclear and mitochondrial genomes[14].

Despite the described advances, critical gaps persist in our understanding of the early steps in mitochondrial translation dynamics. Current models lack temporal resolution to explain how membrane insertion is coordinated with nascent chain synthesis in real time[4,16,17]. The functional importance of ribosome pausing, a hallmark of mitochondrial translation, remains debated: are these pauses regulatory checkpoints for quality control or stochastic inefficiencies[2,14]? Additionally, the mechanisms by which hydrophobic OXPHOS subunits, particularly those with multiple transmembrane domains, are inserted into the IMM remain unclear[4,18]. While structural studies suggest that mtRNC rearrangements may facilitate interactions with OXA1L, kinetic and structural data bridging the process of mRNA translation and membrane insertion are lacking[15].

Here, we addressed the mechanism of cotranslational protein insertion into the IMM with two complementary approaches. We applied ribosome profiling to test the hypothesis that ribosome pausing at specific mRNA positions is linked to the recruitment of the membrane insertase machinery to facilitate membrane integration of transmembrane helices (TMHs). To resolve this, we integrated ribosome profiling to map translation kinetics with cryo-EM snapshots of paused mitochondrial ribosomes. Our data reveal quantitative information on individual mRNA translation and translational pausing events, indicating that the membrane topology of the translation product impacts translation speed. In addition, biochemical purification of native COX1-specific mtRNCs and collection of an extensive cryo-EM dataset comprising more than 50,000 micrographs to overcome low mtRNC density enabled us to resolve structures of specific mtRNC complexes translating COX1 in association with OXA1L/MITRAC. Our structural data reveal cotranslational folding of the COX1 nascent chain within the mitochondrial ribosomal vestibule, correlating with alterations in vestibule dynamics and large-scale changes in mtRNC–OXA1L/MITRAC conformation. These correlated local and global rearrangements modulate nascent chain accessibility and indicate a functional coupling to translation state, as monitored by our ribosome profiling analysis. Thus, our analyses provide insight into how the membrane topology of nascent chains impacts translation speed in mitochondria and support the concept of spatiotemporal coupling between translation and membrane insertion.

## Results

### Investigating mitochondrial translation using ribosome profiling

Mitochondrial-encoded polypeptides are cotranslationally inserted into the inner membrane. The ability to map the positioning of mitochondrial ribosomes on the translated mRNAs represents a powerful strategy to investigate mitochondrial protein biogenesis. Therefore, we established a ribosome profiling approach to address the mechanism of translation-coupled protein insertion. In particular, we adapted the established ribosome profiling and data analysis protocols of Bertolini et al.[19] to meet the requirements for investigation of mitochondrial translation. Specifically, translation in HEK293 cells was arrested by addition of chloramphenicol (CHL) and cycloheximide or by snap-freezing cells in liquid nitrogen. Cell lysates were digested by micrococcal nuclease (MNase) and ribosome populations were separated using sucrose gradient ultracentrifugation. Mitochondrial ribosome fractions were collected and ribosome protected footprints were isolated by phenol–chloroform extraction before library preparation and sequencing (Fig. 1a)[19].

Accurate translatome analysis that provides positional information of ribosomes along the translated mRNA requires efficient and immediate stalling of translation during cell harvest, for example, by antibiotic treatment. To minimize ongoing translation during cell harvest, we first addressed the mitochondrial translation elongation kinetics upon CHL treatment in vivo, by monitoring the production of [$^{35}$S]methionine-labeled translation products at different time points after antibiotic addition. Translation inhibition by CHL stalled translation with fast kinetics. Upon parallel addition of [$^{35}$S]methionine and CHL, mitochondrial translation was efficiently blocked. A full block of translation was apparent after more than 5 min of treatment (Fig. 1b). Therefore, we selected a time of 10 min of treatment for further experiments.

The protocol for ribosome profiling described above enabled enrichment of ribosome footprints, generated by MNase digestion of HEK293 cell lysate. Following MNase treatment, we applied sucrose gradient separation to selectively enrich mitochondrial monosomes from HEK293T cells (Fig. 1c). As the low mitochondrial rRNA content and the low abundance of mitochondrial ribosomes did not allow for detection using 254-nm absorbance, we monitored mitochondrial monosome distribution by western blotting of the gradient fractions. Under the chosen conditions, mitochondrial monosomes were detected in fractions 10–16 of the gradient, above cytosolic monosomes (Fig. 1c).

To support that we faithfully isolated translating monosomes, we performed sucrose gradient centrifugation of MNase-digested mitochondrial lysates after in-organelle translation in the presence of [$^{35}$S]methionine. Both the CHL-treated mitochondria and the control displayed similar separation patterns of RNCs (Fig. 1d and Extended Data Fig. 1a).

To profile translation, footprints of mitochondrial monosomes were isolated from the relevant sucrose gradient fractions and subjected to library preparation and deep sequencing. Upon bioinformatic analysis, we noticed that footprints of different sizes were recovered from the mitochondrial ribosomes (Fig. 1e). These variations of footprints were previously observed for mitochondrial profiling[20–25]. For further analyses, we selected 30–40-nt footprints on the basis of previous studies and available structural data on active ribosomes (Fig. 6)[25]. For our analyses, we adapted the mitochondrial genome to take into account that the transcripts undergo post-transcriptional modification, such as processing and polyadenylation, which generates the STOP codon in case of the transcripts encoding ND1, ND2, ND3, ND4, CYTB, COX3 and ATP6 (ref. 26). In addition, we provided the two bicistronic transcripts (ATP8–ATP6 and ND4L–ND4) as single open reading frames for genome alignment (Extended Data Fig. 1b). Applying a footprint frame of 30–40 nt, all mitochondrial mRNAs were present in our dataset (Fig. 1f,g). Most reads were obtained from transcripts encoding

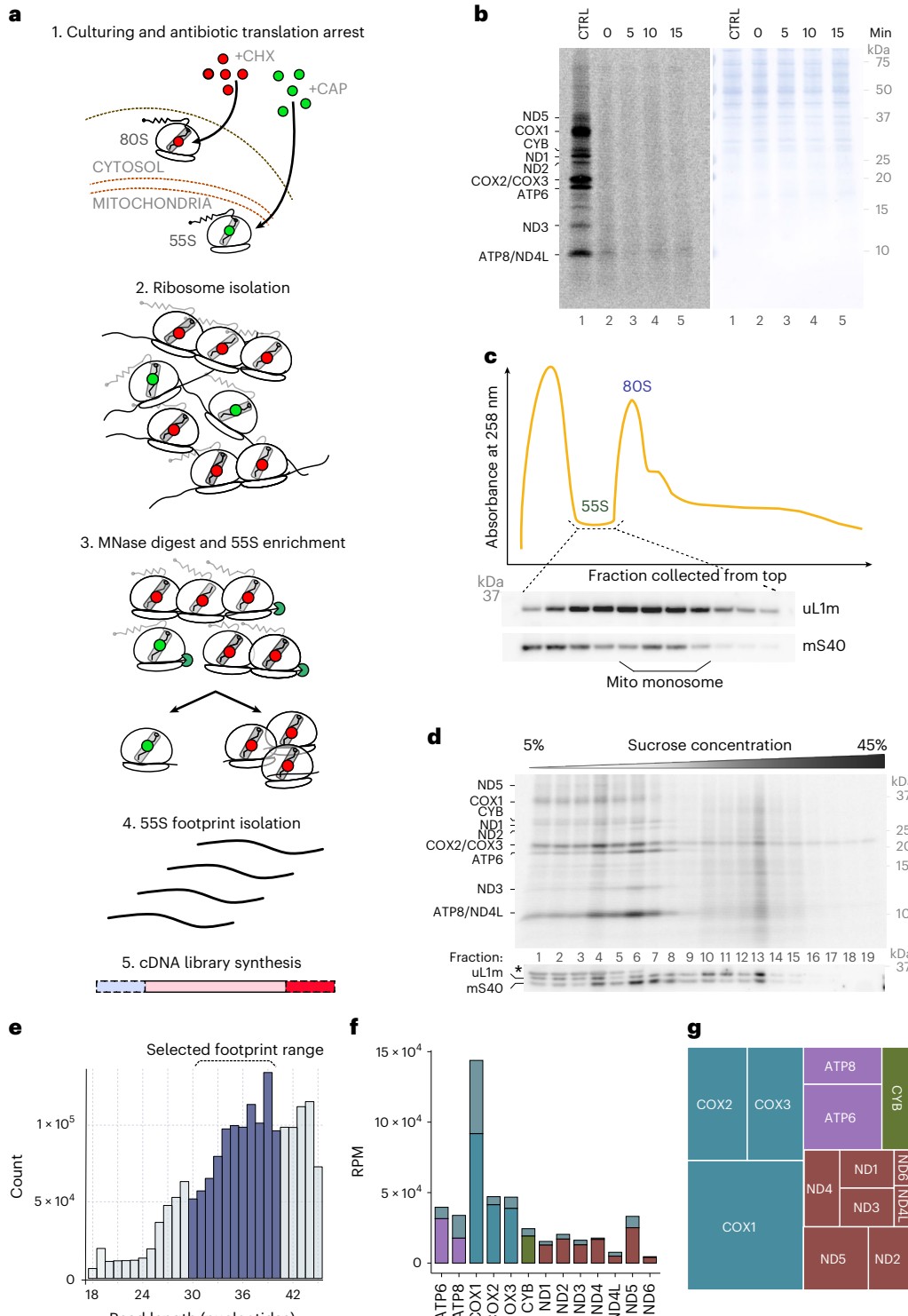

**Fig. 1 | Footprint isolation and profiling of mitochondrial mRNAs. a**, Schematic representation of the procedure of our ribosome profiling approach. **b**, In vivo [35S]methionine labeling of mitochondrial translation products in wild-type cells. Proteins were separated by SDS–PAGE followed by western blotting (right; Coomassie staining) and digital autoradiography (left). Mitochondrial translation was stalled with CHL at indicated time points. The control sample was lacking CHL ($n = 2$). **c**, Absorption profile of average 5–45% sucrose gradient separating mitochondrial and cytosolic ribosomes. Mitochondrial monosome was detected by western blotting with indicated antibodies ($n = 6$). **d**, [35S]methionine labeling of mitochondrial translation products in purified mitochondria. Mitochondrial translation was terminated by addition of CHL, solubilized and MNase-digested. Proteins were separated on a sucrose density gradient, fractionated and analyzed by SDS–PAGE followed by digital autoradiography. The presence of ribosomal subunits was detected by western blotting with indicated antibodies ($n = 3$). **e**, Size distribution of isolated sequenced ribosomal footprints presented as bar graphs (y axis, number of reads for each frame; x axis, footprint size range). **f**, Number of ribosomal footprints isolated from CHL-treated cells for each mRNA represented as reads per million sequenced reads. Gray bar, number of footprints aligning to first 15 codons; colored bars, number of footprints aligning to codon 16 and downstream. (Extended Data Figs. 1 and 2). **g**, Tree map of isolated footprints aligning to mitochondrial transcripts sorted to indicate number of isolated footprints in relation to OXPHOS association: purple, CV; blue, CIV; green, CIII; red, CI ($n = 3$).

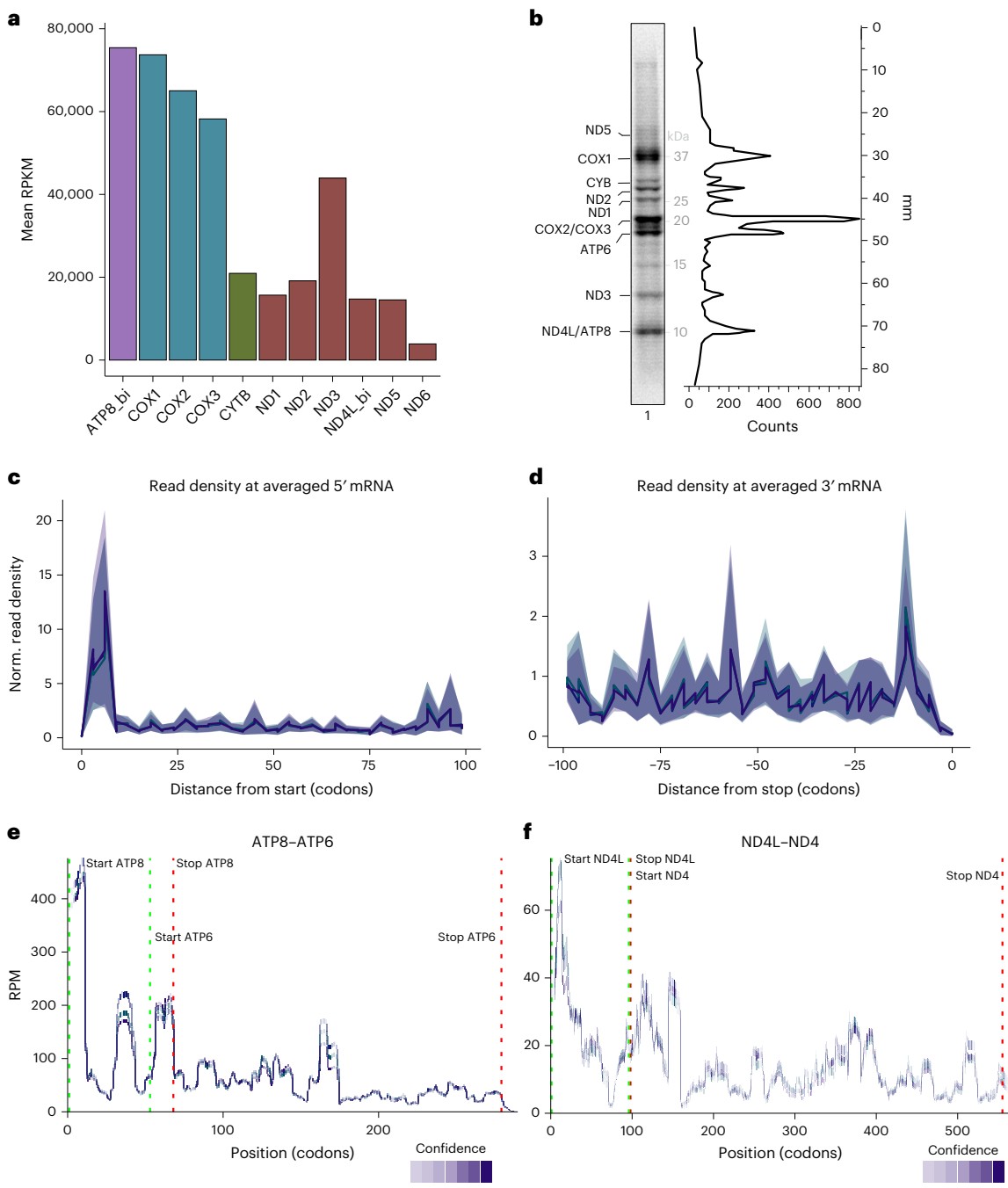

**Fig. 2 | Assessment of mitochondrial translation. a**, Number of ribosomal footprints isolated from each mRNA represented as RPKM. Colored bars indicate OXPHOS complex origin: purple, CV; blue, CIV; green, CIII; red, CI (*n* = 3). **b**, [³⁵S] methionine labeling of mitochondrial translation products in cells analyzed by SDS–PAGE and digital autoradiography (*n* = 3). **c,d**, Mitochondrial metagenes analysis from CHL-treated cells: first 100 codons from the 5′ start (**c**) and 100 codons upstream of the 3′ stop codon (**d**); error is indicated by the shading in line color (*n* = 3). **e,f**, Isolated footprints aligning to ATP8–ATP6 (**e**) and ND4L–ND4 (**f**) obtained from CHL-treated cells (*n* = 3). Confidence is indicated as opacity (Extended Data Fig. 2).

complex IV (CIV) subunits with approximately 49% of all reads (COX1, 20%; COX2, 15%; COX3, 13%). Transcripts encoding subunits of other OXPHOS complexes contributed with 30% (complex I, CI), 5% (complex III, CIII) and 17% (complex V, CV) (Fig. 1f,g). Read counts of individual transcripts normalized by library size and transcript length, expressed as reads per kilobase of transcript per million mapped reads (RPKM), resembled approximately the translation profile observed upon [³⁵S] methionine labeling of mitochondrial translation products (Fig. 2a,b).

To reveal common patterns of read abundances among transcripts, we aligned all mRNAs at their start or their stop codon and averaged read densities of all mRNAs creating a metagene ribosome distribution profile that represents all mitochondrial translation events. Figure 2c,d reveals ribosome read densities downstream of the 5 start codon (Fig. 2c) and upstream of the 3′ stop codons (Fig. 2d). Increased read densities were found at the 5′ start codon, spanning an average distance of ten codons (Fig. 2c). Reads also accumulated toward the 3′ stop codon of transcripts, whereas, shortly upstream of the stop codon, a drop in read density was observed (Fig. 2d).

Metagene profiles showed ribosome enrichment -10 codons after the start (Fig. 2c) and before the stop of the drop (Fig. 2d). Individual

transcripts (Extended Data Fig. 2a–m) confirmed strong initiation pausing, strongest for COX1 (37% reads in first ten codons); bicistronic overlaps were noted for ATP8–ATP6, ND4L–ND4 (Fig. 2e,f). The 3′-end read reduction suggests accelerating elongation. Thus, 30–40-nt footprints with a modified genome enable precise mitochondrial translation assessment.

## In organello silencing affects ribosome occupancy on target mRNAs

Translation of selected mRNAs can be blocked by importing a precursor–morpholino chimera into purified mitochondria. For this, chimeras consisting of a mitochondrial precursor protein (Jac1) fused to a polymorpholino directed against a mitochondrial mRNA are imported into mitochondria where the morpholino interacts with the cognate transcript to block translation[17]. We reasoned that combining the ribosome profiling approach with morpholino-based knockdown (KD) is ideally suited to address the mechanism of translational silencing while also validating the technical quality of the pipeline. Therefore, we imported chimera directed against either COX1, COX2 or ND2 mRNA into purified mitochondria. We used morpholinos targeting nucleotides 1–19 of COX1, 1–23 of COX2 or 12–29 of ND2 mRNAs. Previous data indicated that morpholino chimeras efficiently reduced the synthesis of the corresponding newly synthesized polypeptide without significantly changing the expression of other mRNAs[17]. We performed ribosome profiling from mitochondria purified under the chimera-treated conditions. The presence of chimera in mitochondria reduced the expression levels displayed in RPKM of COX1 and COX2 mRNAs by 70% and 60%, respectively, when compared to the control sample (Fig. 3a). At the level of newly synthesized polypeptides, the chimera treatment efficiently reduced the amount of newly made COX1 and COX2 proteins[17]. For both, COX1 and COX2, the use of the respective morpholinos resulted in a reduction of ribosome densities throughout the transcript of the target mRNA (Fig. 3e,f). The presence of COX1[1–19] chimera in mitochondria resulted in higher normalized read counts on all other mRNAs; however, we cannot rule out that this was a consequence of the relative reduction of normalized reads on COX1 (Fig. 3d,f and Extended Data Fig. 3). On the other hand, the presence of COX2[1–23] resulted in more complex changes in translation. Here, total normalized reads on most other transcripts were reduced, with the exception of COX1, COX3 and ND2, which displayed the opposite effect. Curiously, we observed an increased ribosome abundance close to the start codon for most of the other transcripts not targeted by the COX2[1–23] morpholino and reduced ribosome density downstream of this region (Fig. 3c,i). This effect was especially pronounced for COX1 when COX2 translation was blocked, where the overall read density increased and the densities of reads located downstream of codon 15 were falling below control (untreated) levels toward the 3′ end, with the exception of reads aligning close to the stop codon (Fig. 3c). In summary, the ribosome profiling data support that chimera treatment reduces target translation in mitochondria. Interestingly, in the case of blocked COX2 translation, effects on translation of other mitochondrial transcripts were observed. This finding agrees with previous radiolabeling analyses of mitochondrial translation products in the presence of COX2 chimera[17]. Similarly, ND2[12–29] morpholino treatment reduced ribosome occupancy on the ND2 transcript (Fig. 3a,d,g,j), serving as a CI control that behaved comparably to COX2 silencing without additional inter-mRNA effects We conclude that the established pipeline faithfully provides information on mitochondrial gene expression.

## Identification of translational pausing by ribosome profiling

Work on COX1 translation identified an RNC complex paused at a stage when the nascent COX1 is partially inserted into the IMM[2]. In the membrane, the nascent COX1 is associated with the protein insertase OXA1L and biogenesis factors such as the COX1-specific assembly factor C12ORF62 (COX14) and MITRAC12 (COA3)[2,13,17,27]. Upon translation

of mitochondrial-encoded polypeptides in purified mitochondria, translation intermediates of COX1 could be detected, which were lost in the presence of chimera affecting specifically the COX1 translation (Fig. 4a, lane 2). As compared to previously visualized translation intermediates, we were able to resolve two additional fragments: one migrating slower than ND3 and one early translation product migrating faster than ND4L and ATP8 (Fig. 4a). Translation intermediates of COX1 were efficiently coimmunopurified with C12ORF62[FLAG] (Fig. 4b). To define translation intermediates of COX2, we treated mitochondria with COX2 chimera to specifically block COX2 translation. Indeed, we identified a prominent specific translation intermediate of COX2 that was lost upon COX2 silencing (Fig. 4a, lane 4). While COX1 spans the inner membrane with twelve transmembrane domains and exposes both termini to the matrix, COX2 exposes C and N termini into the intermembrane space while spanning the membrane twice. Moreover, translation intermediates of the CI subunit ND2, which were sensitive to treatment with ND2[12–29] morpholino (Extended Data Fig. 5), were efficiently coimmunopurified with MITRAC15[FLAG] (COA1) (Fig. 4c, lanes 3 and 4). In addition, we observed a specific translation intermediate of ATP8 of 3.6 kDa (ref. 28) that was sensitive to chimera treatment (Extended Data Fig. 1c).

Profiling data of the respective mRNAs COX1, COX2 and ND2 presented translational pausing events corresponding to the identified fragments (Fig. 4d). The distribution of ribosomes on the COX1 mRNA showed expected pausing sites, for example, at codons 180 and 230 (Fig. 4d, left). COX2 had a lower number of pausing sites. Interestingly, one major pausing site fell onto codons 92/93 correlating to the tunnel emergence of the second TMH (Fig. 4d, middle). On the contrary, ND2 mRNA translation showed several sites with increased read amounts, particularly three major accumulations at codons 92/93, 144/145 and 305/306, corresponding closely to the identified fragments. Unfortunately, the two early translation intermediates of ND2 were of similar size as ND3 and ND4L and could, therefore, not be identified in the radiolabeling analysis (Fig. 4c, light-red circles). Lastly, the observed ATP8 fragment corresponded to increased reads at codons 35–38 (Fig. 2e).

Applying C12ORF62 (COX14) immunoprecipitation, nascent chain fragments of COX1 have been identified and estimated in their length[2,17,27]. All of these identified COX1 fragments—with estimated amino acid lengths of below 170, 212 and 280—matched areas with an increased accumulation of reads. Investigation on codon level revealed that the fragments f3 (280 aa) and f2 (212 aa) aligned directly with paused codons in the P-site (Fig. 4d). The third fragment f1 (~170 aa) fell between two minor pausing events. Yet, the fragment length estimation by SDS–PAGE is error-prone because of the properties of the hydrophobic polypeptide and, therefore, cannot be perfectly linked to fragments identified with codon precision in the profiling approach. Next, we aligned the transmembrane domains (black) of COX1 and COX2 and their connecting loop regions (blue, COX1; dark orange, COX2) to the moment of their decoding (Fig. 4d). A high number of reads aligned to early codons of TMHs (black), suggesting translation pausing, in agreement with observed translation fragments[2]. Further toward the 3′ ends of the mRNAs, the pausing sites no longer displayed a notable overlap with the start of TMH translation (Fig. 4d). In the case of the COX2 mRNA, notable ribosome pausing was observed for codons 92/93, in agreement with the size of the COX2 chimera-sensitive fragment identified in in organello translation (Fig. 4a). Similar to COX1 and COX2 translation profiles, the translation profile of ND2 displayed read accumulations corresponding to fragments identified by in organello translation. Two small identified fragments strongly correlate to pausing at codons 92/93 and 144/145 (light red) and larger fragments matching less abundant read pausing sites close to codon 300 and one major site at codon 306 (Fig. 4c,d). A common feature of the nascent chains of all three mRNAs was their correlation of pausing events with the emergence of a topological region linking two TMHs (from now on,

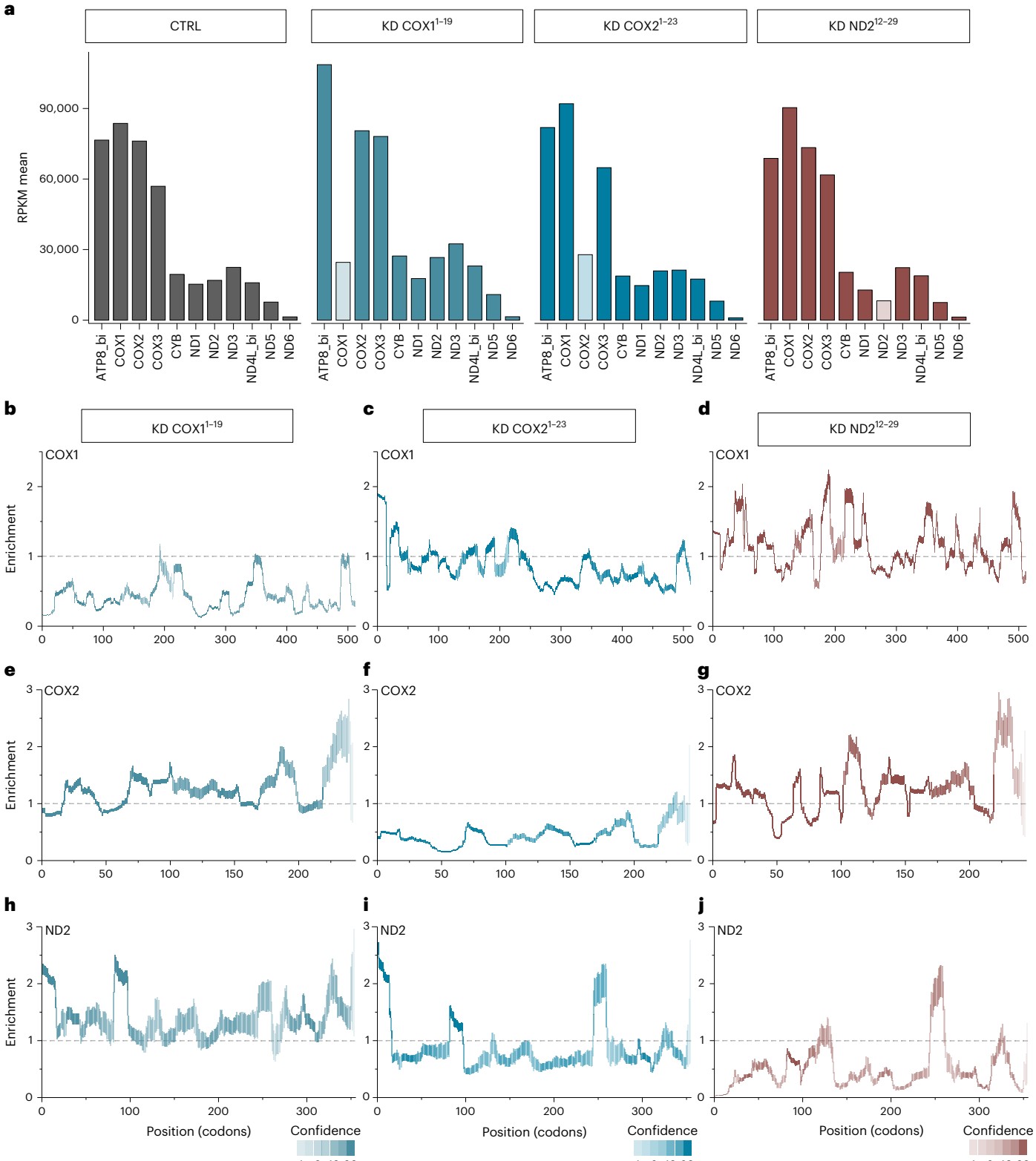

**Fig. 3 | Profiling upon in organello silencing. a**, Purified mitochondrion were subjected to chimera-mediated (KD COX1[1–19], KD COX2[1–23] and KD ND2[12–29]) silencing or left untreated. The number of ribosomal footprints isolated from each mRNA is represented as RPKM ($n = 3$). Gray, control sample; blue, KD COX1[1–19]; aquamarine, KD COX2[1–23]; red, ND2[12–29]. **b,e,h**, Footprint isolations visualized as enrichment over control plots of the indicated mRNAs upon KD COX1[1–19] silencing (one representative replicate; $n = 3$; Extended Data Fig. 3) **c,f,i**, Footprint isolations visualized as enrichment over control plots of the indicated mRNAs upon KD COX2[1–23] silencing (one representative replicate; $n = 3$; see Extended Data Fig. 4) **d,g,j**, Footprint isolations visualized as enrichment over control plots of the indicated mRNAs upon KD ND2[12–29] silencing (one representative replicate; $n = 3$; Extended Data Fig. 5).

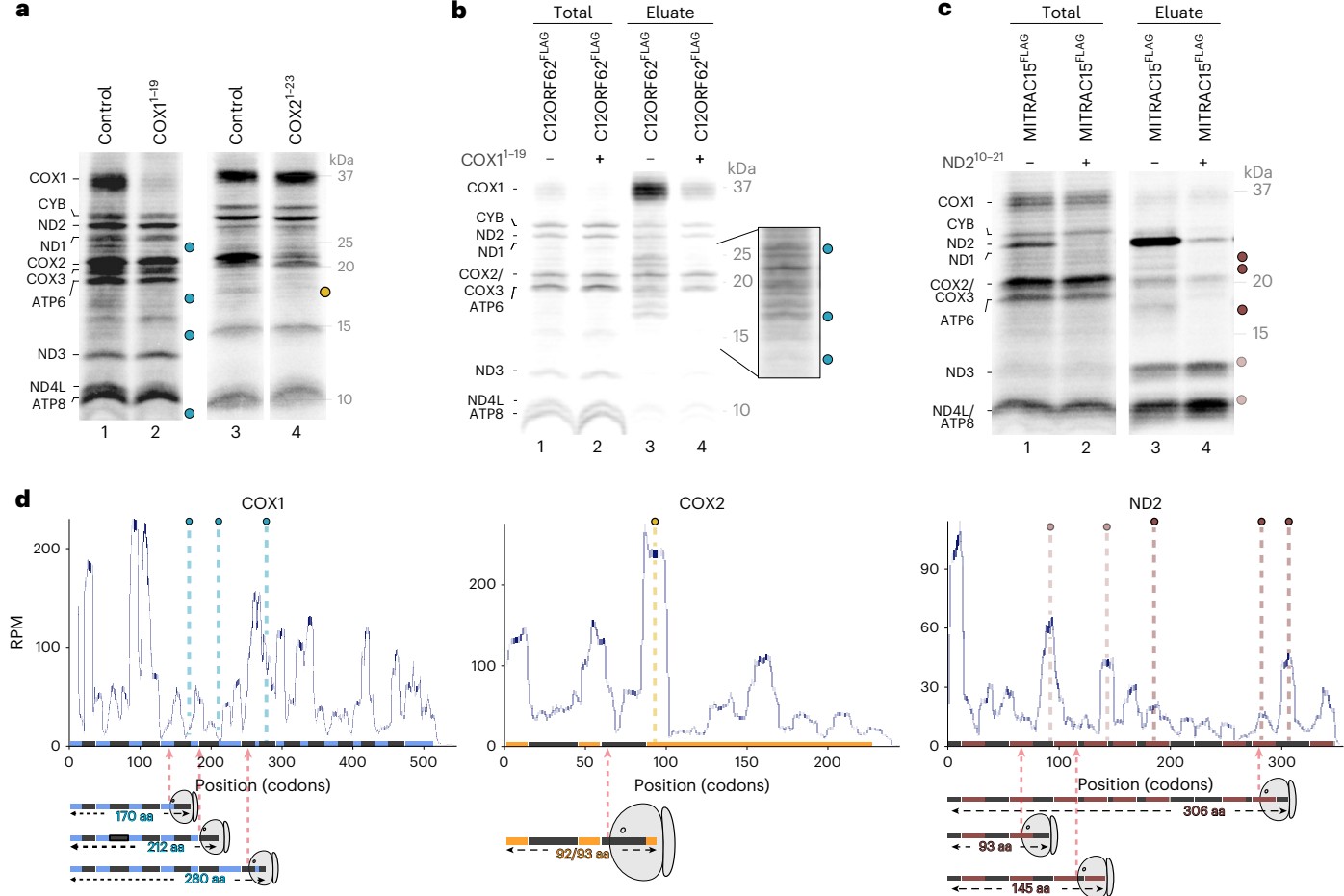

**Fig. 4 | Ribosome pausing correlates with TMH incorporation. a,** [35S] methionine labeling was performed in purified mitochondria in the presence or absence of indicated morpholino chimera targeting COX1 (blue circles) or COX2 (light green). Samples were analyzed by SDS–PAGE, followed by digital autoradiography (n = 3; profiling results in Fig. 3 and Extended Data Figs. 3 and 4). **b,** [35S]methionine labeling was performed in purified mitochondria in the presence or absence of COX1[1–19] morpholino. Samples were subjected to coimmunoisolation using C12ORF62[FLAG] as bait. Samples were analyzed by SDS–PAGE, followed by digital autoradiography (n = 3). **c,** [35S]methionine labeling was performed in purified mitochondria in the presence or absence

of ND2[12–29] morpholino. Samples were subjected to imunoisolation using MITRAC15[FLAG] (COA3) as bait. Samples were analyzed by SDS–PAGE, followed by digital autoradiography (n = 2). **d,** Representative footprint alignment profiles of COX1, COX2 and ND2 mRNA. Dotted lines in blue, yellow and red correspond to polypeptide fragments of COX1 COX2 and ND2 identified in **a**–**c**, respectively. Bars indicate structural features of polypeptides. Dark gray, TMHs; blue, orange and red, topological regions. Cartoons to visualize differences in pausing on codons and fragment length are indicated with red arrows in mRNA profiles (one representative replicate; n = 3; Extended Data Fig. 2–5).

only topological region) from the ribosomal exit tunnel or vestibule (Fig. 4d, red dotted arrows). In summary, the ribosome profiling data recapitulated translation intermediates of COX1 and COX2 observed at the protein level.

**Protein topology links to decoding speed**

All human mitochondrial translation products represent proteins that span the inner membrane. On the basis of their topology, these proteins can be classified into two groups, a first one in which the N terminus faces the matrix (N-in) and a second in which the N terminus faces the intermembrane space (N-out) (Fig. 5b). We reasoned that cotranslational insertion of the polypeptide chain should present different challenges to the process depending on the topology of the polypeptide's N terminus. While proteins with N-out topology can be directly inserted into the lipid phase (Fig. 5a, right; ND1), proteins with N-in topology must undergo a topology switch to generate a loop and maintain the N terminus in the matrix (Fig. 5a, left; COX3). On the basis of the profiles of mRNAs with N-in topology, we speculated that the establishment of a correct nascent chain topology in the context of the environment provided by exit tunnel, membrane and

hydrophobicity of the TMHs impacted translation elongation rates. To address whether translation in mitochondria was linked to the insertion process, we categorized all proteins depending on their topology, aligned them at the beginning of their first TMH and performed metagene analysis (Fig. 5c). This analysis enabled us to synchronize positional footprint data downstream of the decoding start of the first TMH. When applying a minimal averaged density threshold of 200, on average, two major ribosome density accumulations and pauses were present near codon ~30 (TMH N terminus arrives at vestibule) and a second at codon ~78 (second TMH N terminus arrives at vestibule) for N-in and N-out proteins (Fig. 5c,d; I–IV). For N-in polypeptides, one additional accumulation could be resolved at codon ~15, whereas, for N-out polypeptides, one additional accumulation could be resolved at codon ~55 (Fig. 5a,c,d). Interestingly, the accumulations identified at positions 15 (N-in), 30, 55 (N-out) and 78 repeat after roughly 50 codons in a similar pattern for both groups. Yet, for the N-out group, the density accumulation downstream of codon 50 was increased. Accordingly, translation of mRNAs encoding the two classes of proteins with different topologies appear to differ. On the basis of our data, we concluded that alterations in the topology of the newly synthesized

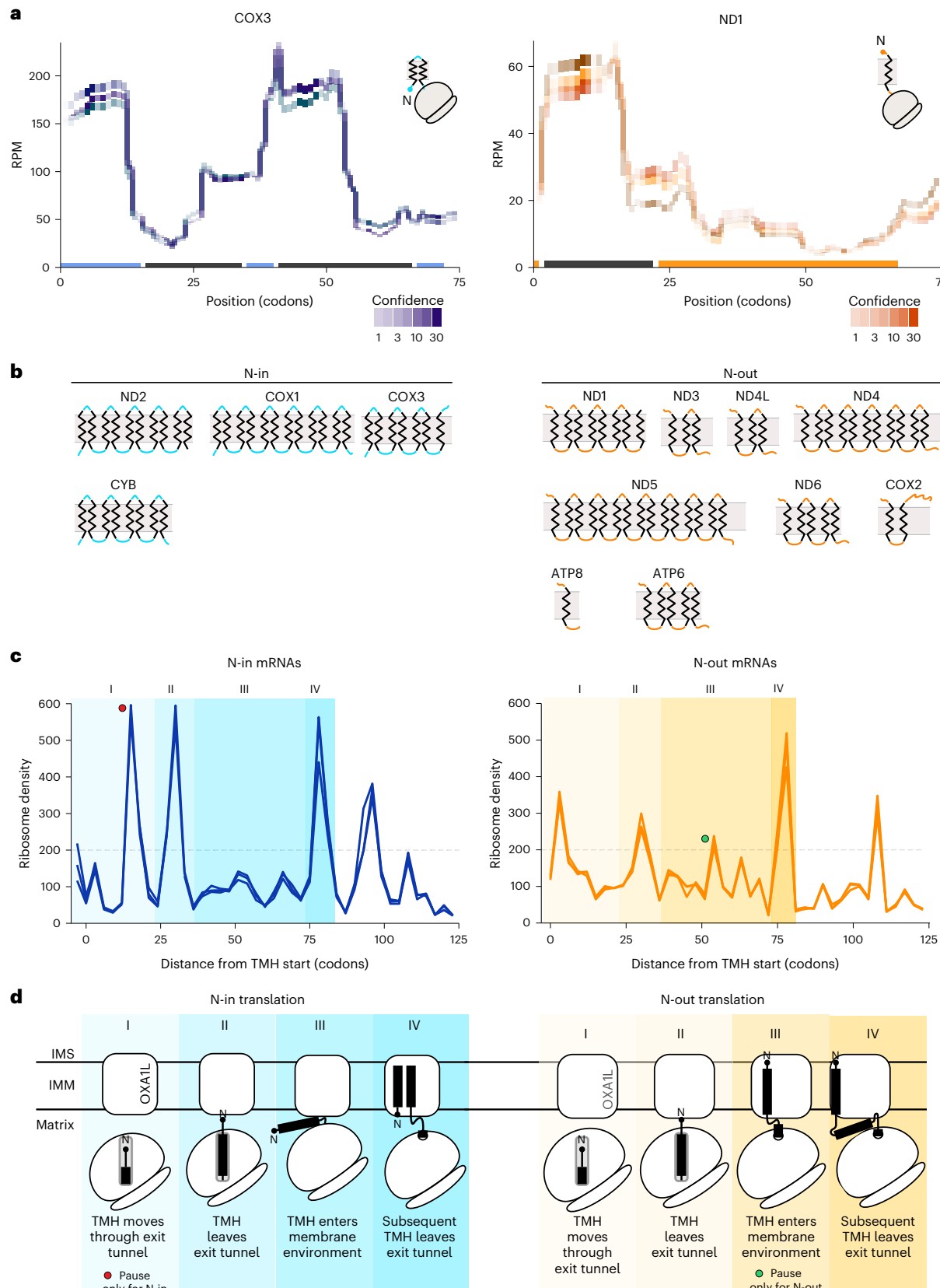

**Fig. 5 | Translation speed adapts depending on insertion of TMH. a**, Read alignment of COX3 (left; blue) and ND1 (right; orange) as representative mRNAs with N-in and N-out topology (*n* = 3). Bars indicate the structural features of transcripts. Dark gray, TMHs; blue, topological region (Extended Data Fig. 2). **b**, Schematic presentation of translation products in the inner membrane. **c**, Metagene alignments for mRNAs aligned at the first codon encoding the first TMH filtered into groups on the basis of N terminus topology. Blue, N terminus retained in matrix (left; N-in); orange, N terminus translocated to IMS (right; N-out). Normalized ribosome densities are displayed on the *y* axis of three biological replicates (*n* = 3). **d**, Schematic representation of TMH insertion events identified in **c**. Left, insertion of N-in proteins (blue). Right, insertion of N-out proteins (yellow). Matching events in **c**,**d** are denoted by similar background coloration.

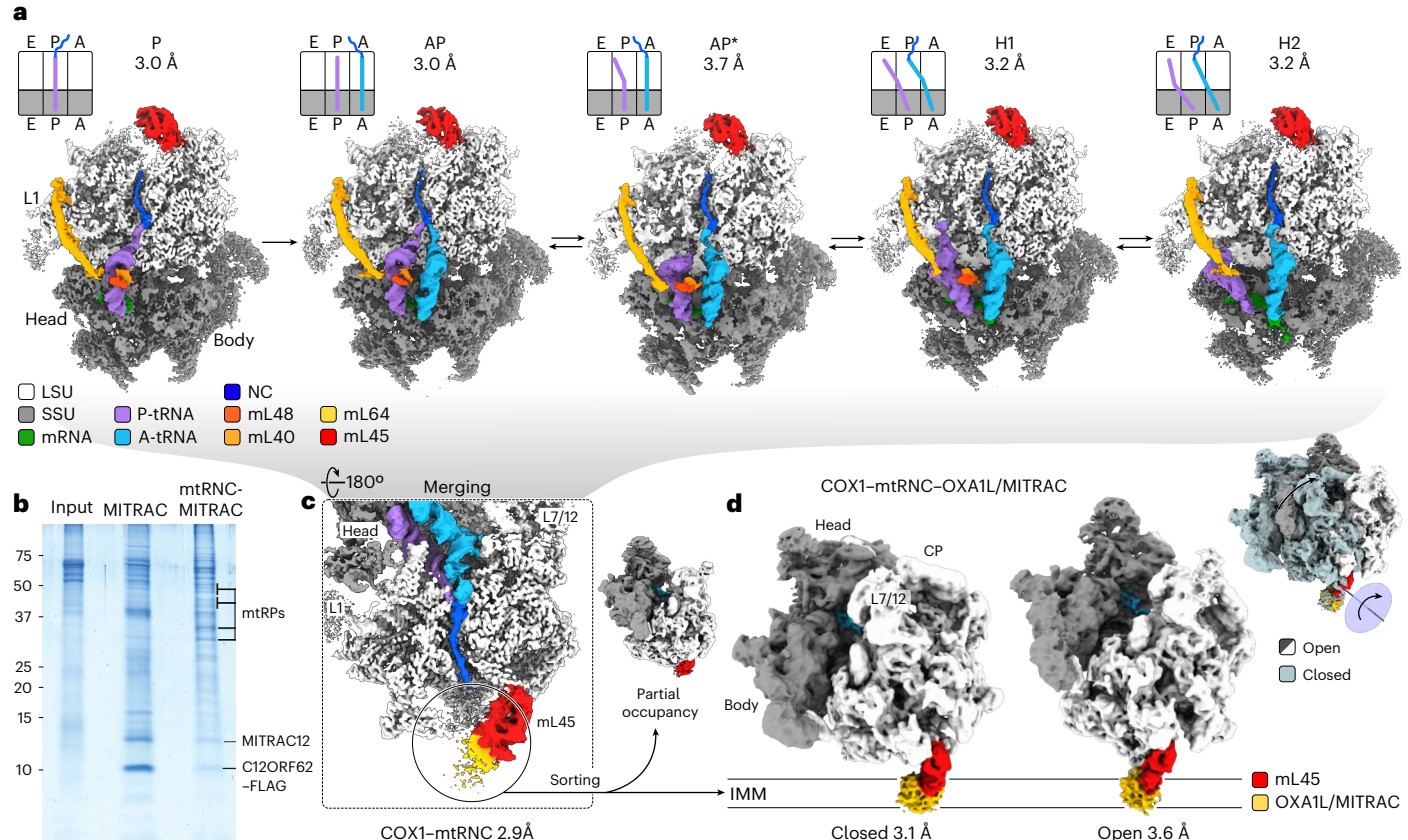

**Fig. 6 | Structure and dynamics of human COX1-translating mtRNC–MITRAC complexes. a**, Cryo-EM maps of COX1–mtRNCs in different states of translation, shown as cross-sections (from left to right): post-translocation state (P), classic pretranslocation state (AP), intermediate state (AP*) and tRNA hybrid states (H1 and H2). A, P and E, tRNA binding sites; NC, COX1 nascent chain; mL, proteins of mtLSU; L1, L1 stalk of LSU; head and body, head and body of SSU. **b**, Purification of COX1-translating mtRNCs for cryo-EM using C12ORF62 as bait. MITRAC COX1 assembly intermediate and mtRNC–OXA1L/MITRAC were purified by C12ORF62[FLAG] immunoisolation and analyzed by SDS–PAGE and Coomassie staining (*n* = 1). mtRPs, mitoribosomal proteins; C12ORF62 and MITRAC12, assembly factor-components of the MITRAC complex[2,14]. **c**, Average cryo-EM map of COX1–mtRNC (close-up view). Note the scattered density at the tunnel exit (circle) indicating incomplete occupancy with membrane-integral OXA1L/MITRAC. **d**, COX1–mtRNC–OXA1L/MITRAC complexes sample two distinct conformational states. Left and middle, cryo-EM maps of COX1–mtRNC–OXA1L/MITRAC complexes in closed and open states obtained by sorting of cryo-EM data in exit area (circle in **c**), lowpass-filtered to 6-Å resolution for clarity. Top right, a rotational movement opens up the region between membrane-integral OXA1L/MITRAC and mtRNC. IMM, schematic of IMM indicating approximate orientations; CP and L7/12, central protuberance and L7/12 stalk of the LSU, respectively.

polypeptide chain are imprinted in the translation process probably because of the requirement of hairpin loop formation before membrane insertion (Fig. 5c,d).

**Cryo-EM of COX1-translating RNC–OXA1L/MITRAC complexes**

To assess how translation is coupled to membrane insertion in mitochondria at the molecular level, we visualized COX1-translating mitochondrial ribosome–nascent chain complexes by cryo-EM (COX1–mtRNC) (Figs. 6 and 7). In mitochondria, COX1 represents the most translated transcript; therefore, COX1–mtRNC complexes are expected to represent the largest population of mtRNCs (Fig. 2a). In the COX1–mtRNC complexes, the partially membrane inserted nascent chain is in complex with assembly factors C12ORF62 and MITRAC12, as well as the OXA1L insertase and probably other membrane proteins such as TMEM126A (ref. 18). Therefore, we refer here to the complexes as COX1–mtRNC–OXA1L/MITRAC complexes. Complexes were prepared for cryo-EM using a FLAG-tagged version of C12ORF62 (C12ORF62[FLAG]), which allows selective purification of COX1–mtRNC–OXA1L/MITRAC (Figs. 6b and 7a and Methods). In contrast to our previous biochemical work, we used the detergent PCC (4-*trans*-(4-*trans*-propylcyclohexyl)-cyclohexyl α-maltoside) instead of digitonin to extract complexes from purified mitochondria.

By sorting cryo-EM data according to tRNA states, we obtained five structures of mtRNCs in distinct states of translation at 3.0–3.7-Å resolution (Fig. 6a, Extended Data Fig. 6 and Table 1). These structures capture the stepwise movement of tRNAs through the mitochondrial ribosome[29], starting with the nonrotated post-translocation state (P), in which the tRNA occupies the classical P/P position (P-site on both small and large ribosomal subunits, mtSSU and mtLSU). This is followed by the classical pretranslocation state (AP), where tRNA binding results in A/A and P/P configurations, followed by tRNA hybrid states (H1 and H2). Transition into hybrid states is driven by mtSSU rotation, upon which the tRNAs on the mtLSU into A/P and P/E configurations. In the H1 state, the deacylated tRNA remains bound to the P-site on the mtSSU, while its elbow adopts an intermediate position between P and E sites on the mtLSU (P/E*; Extended Data Fig. 6d), distinct from the canonical P/E position in H2 observed previously[29]. Notably, we identify a transient intermediate of tRNA translocation in eukaryotes (AP*) that bridges the classical AP state and the initial H1 hybrid state. To the best of our knowledge, comparable intermediates have only been described in bacterial translation at low resolution ('Pre2' in a previous study[30]). In the AP* state, the nascent chain-bound tRNA moves into an A/P configuration, while the deacylated P-site tRNA retracts its CCA-end from the peptidyltransferase center (PTC) on the mtLSU

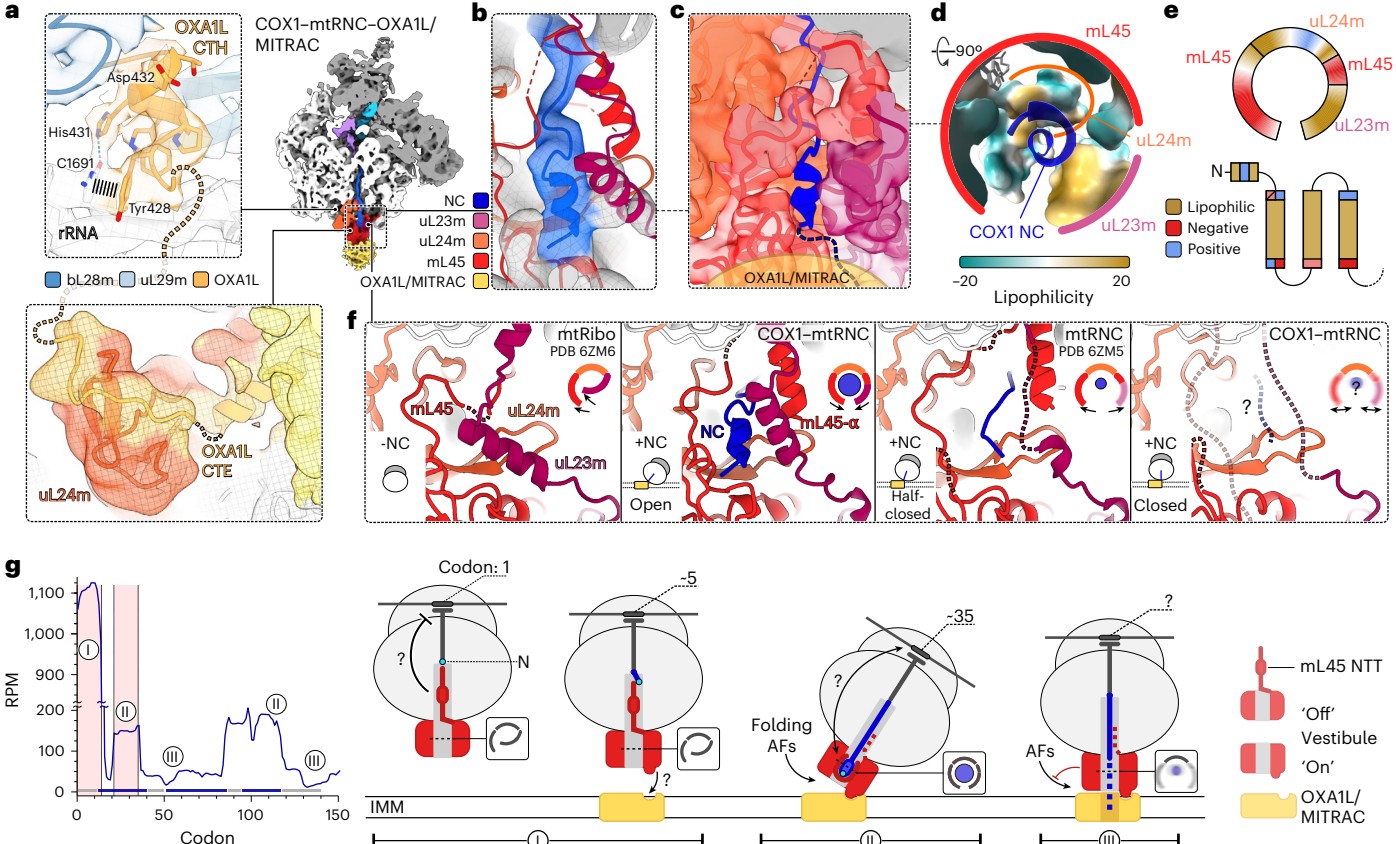

**Fig. 7 | Cotranslational folding at the tunnel exit in COX1–mtRNC.**
**a**, Interactions of insertase OXA1L with mtLSU in COX1–mtRNC–OXA1L/MITRAC. Left, close-up views of OXA1L contact sites in the open state. Semitransparent mesh, experimental densities filtered to local resolution; CTH and CTE, C-terminal helix and extension of OXA1L, respectively. Right, overview.
**b**, Experimental density for vestibule region in open COX1–mtRNC–OXA1L/MITRAC, indicating cotranslational folding of the COX1 nascent chain (blue). Semitransparent mesh, cryo-EM density lowpass-filtered to 6 Å and rendered at high threshold (5σ). **c**, Nascent chain path in the open COX1–mtRNC–MITRAC. Close-up view of tunnel exit with COX1 nascent chain folding into an α-helical element in the vestibule formed by uL23m, uL24m and mL45. The dashed line indicates the potential path of the nascent chain towards OXA1L/MITRAC. Molecular model surfaces are shown in semitransparent representation.
**d**, Ultrawide view of the amphipathic vestibule in the open state colored according to lipophilicity potential, shown from top. **e**, Schematic comparison of the amphipathic vestibule (top) and the amphipathic properties of COX1 N-terminal α-helix (α0) and TMH1–TMH3 (bottom). **f**, Nascent-chain-dependent dynamics of the vestibule region. From left to right: mitochondrial ribosome structure in absence of nascent chain (mtRibo; Kummer et al.[32] and Itoh et al.[4]), COX1–mtRNC–MITRAC in the open state, mtRNC–OXA1L complex with unspecified nascent chain Itoh et al.[4] and COX1–mtRNC–MITRAC in the closed state. In the closed COX1–mtRNC, mL45 NTT, uL23 linker region and COX1 nascent chain are unresolved in the exit region ('?'), indicating high flexibility. mL45α, α-helix of mL45 forming in presence of nascent chain[4,32]. Left insets, cartoons illustrating orientation of mtRNCs relative to membrane. Right insets, schematic cross-sections of vestibule, seen from top. **g**, Proposed model for coupling of translational stalling with mtRNC structural rearrangements. The plot shows ribosome footprint data of early COX1 translation (average of replicates under cryo-EM conditions). I–III denote different stages of translation. For stage I, schematics depict mL45 blocking the exit tunnel before the onset of translation (left; Kummer et al.[32] and Itoh et al.[4]) and during early translation (right; Koripella et al.[44]), which may contribute to initial ribosome pausing, providing a window for ribosome docking to OXA1L/MITRAC at the membrane. In stage II, emergence of the COX1 N-terminal α-helix in the vestibule coincides with a second pausing event, which may be facilitated by nascent chain interactions with the vestibule. The opening of the COX1–mtRNC versus OXA1L/MITRAC may create space for binding of auxiliary factors (AFs) and, at longer chain lengths, for cotranslational folding. In stage III, closure of COX1–mtRNC onto the membrane impedes access of AFs from the matrix and may facilitate unhindered translation and nascent chain integration into the membrane by OXA1L/MITRAC. Insets, schematic cross-sections of vestibule. 'Off' and 'on' denote mL45 conformations that may disfavor or favor OXA1L binding, respectively, as discussed in the text.

(Extended Data Fig. 6d). Mitochondria-specific proteins mL40, mL48 and mL64 contribute to stabilizing these intermediate states (Fig. 6a and Extended Data Fig. 6d). mL40 and mL48 bridge between A-site and P-site tRNAs specifically in the AP* and H1 states, while mL64 interacts with the elbow of the P-site tRNA throughout translocation[29]. Notably, the AP* state is sampled approximately tenfold more frequently than the analogous Pre2 state in bacteria (Extended Data Fig. 6a)[30], suggesting that the mitochondria-specific proteins may promote efficient hybrid-state formation. All five mtRNC structures display clear density for a nascent chain within the exit tunnel, most likely corresponding to COX1 translation intermediates, as we purified complexes using the COX1-specific MITRAC-component C12ORF62 (ref. 2).

To resolve the ribosome-associated membrane-integral OXA1L/MITRAC complex, the cryo-EM data representing distinct tRNA states were merged and sorted for the region at the tunnel exit. In this way, we were able to capture two states of the COX1–mtRNC–OXA1L/MITRAC complex, 'closed' and 'open' at 3.1-Å and 3.6-Å overall resolution (Fig. 6c,d, Extended Data Fig. 6, Table 1 and Methods). Both states show an additional large density of similar shape and size but different orientation, extending beyond the ribosomal tunnel exit, which correlates with the expected position of the OXA1L/MITRAC complex in the IMM. Whereas the local resolution of this additional density largely ranges from only 6 to 15 Å (<4 Å for C-terminal helix of OXA1L), combining our structural and biochemical data identifies this density as the

## Table 1 | Cryo-EM data collection, refinement and validation statistics

| Ribosomal complex | P | AP | AP* | H1 | H2 | Open | Closed |
|---|---|---|---|---|---|---|---|
| **EM Data Bank and PDB identifiers** | EMD-55836 | EMD-55837 | EMD-54637 PDB 9S7B | EMD-54638 PDB 9S7C | EMD-55838 | EMD-54639 PDB 9S7D | EMD-54640 PDB 9S7E |
| **Data collection and processing** | | | | | | | |
| **Microscope** | Titan Krios | Titan Krios | Titan Krios | Titan Krios | Titan Krios | Titan Krios | Titan Krios |
| Camera | Falcon III | Falcon III | Falcon III | Falcon III | Falcon III | Falcon III | Falcon III |
| Magnification | 59,000 | 59,000 | 59,000 | 59,000 | 59,000 | 59,000 | 59,000 |
| Voltage (kV) | 300 | 300 | 300 | 300 | 300 | 300 | 300 |
| Electron exposure (e⁻ per Å²) | 30–45 | 30–45 | 30–45 | 30–45 | 30–45 | 30–45 | 30–45 |
| Defocus range (µm) | 0.2–2 | 0.2–2 | 0.2–2 | 0.2–2 | 0.2–2 | 0.2–2 | 0.2–2 |
| Pixel size (Å) | 1.16 | 1.16 | 1.16 | 1.16 | 1.16 | 1.16 | 1.16 |
| Symmetry imposed | $C_1$ | $C_1$ | $C_1$ | $C_1$ | $C_1$ | $C_1$ | $C_1$ |
| Initial particles (no.) | 2,068,820 | 2,068,820 | 2,068,820 | 2,068,820 | 2,068,820 | 2,068,820 | 2,068,820 |
| Final particles (no.) | 102,321 | 72,838 | 15,253 | 40,778 | 38,389 | 23,725 | 70,865 |
| Map resolution (Å) | 3.0 | 3.0 | 3.7 | 3.2 | 3.2 | 3.6 | 3.1 |
| FSC threshold | 0.143 | 0.143 | 0.143 | 0.143 | 0.143 | 0.143 | 0.143 |
| Map resolution range (Å) | 2.9–20 | 2.9–20 | 3.5–20 | 3.0–20 | 3.0–20 | 3.5–20 | 3.0–20 |
| **Refinement** | | | | | | | |
| Initial models used (PDB code) | - | - | 7QI5 | 7QI5 | - | 7QI5, 6ZM5 | 7QI5, 6ZM5 |
| Model resolution§ (Å) | - | - | 5.3 | 3.8 | - | 4.6 | 3.6 |
| FSC threshold | - | - | 0.5 | 0.5 | - | 0.5 | 0.5 |
| Model resolution range (Å) | - | - | 5.3–6 | 3.8–6 | - | 4.6–6 | 3.6–6 |
| Map sharpening B factor (Å²)◊ | - | - | N/A | N/A | - | N/A | N/A |
| Model composition: nonhydrogen atoms/residues/RSCC¶ | | | | | | | |
| Total | - | - | 179,748/17,885/0.75 | 180,036/17,893/0.80 | - | 180,789/17,991/0.77 | 180,448/17,942/0.79 |
| RNA | - | - | 58,405/2,752/ 0.81 | 58,496/2,757/ 0.85 | - | 58,489/2,756/ 0.82 | 58,496/2,757 /0.83 |
| Proteins | - | - | 121,169/14,876/0.74 | 121,169/14,876/0.79 | - | 121,922/14,974/0.76 | 121,579/14,923/0.78 |
| Cumulative RSCC (%)>0.8/>0.6/>0.4 | - | - | 0.53/0.85/0.95 | 0.67/0.90/0.97 | - | 0.57/0.86/0.95 | 0.64/0.89/0.96 |
| *B* factors (Å²) | | | | | | | |
| Protein | - | - | 263.77 | 243.04 | - | 252.62 | 216.39 |
| Nucleotide | - | - | 251.54 | 205.78 | - | 228.47 | 184.99 |
| Ligands and ions | - | - | 191.5 | 176.32 | - | 183.24 | 149.64 |
| Root-mean-square deviations | | | | | | | |
| Bond lengths (Å) | - | - | 0.005 | 0.007 | - | 0.006 | 0.006 |
| Bond angles (°) | - | - | 0.921 | 0.930 | - | 1.126 | 1.142 |
| **Validation** | | | | | | | |
| MolProbity score | - | - | 0.66 | 0.66 | - | 0.91 | 0.78 |
| Clashscore | - | - | 0.45 | 0.46 | - | 1.39 | 0.93 |
| Rotamer outliers (%) | - | - | 0.00 | 0.00 | - | 0.02 | 0.02 |
| Ramachandran plot | | | | | | | |
| Favored (%) | - | - | 98.18 | 98.16 | - | 97.85 | 98.04 |
| Allowed (%) | - | - | 1.77 | 1.79 | - | 2.14 | 1.94 |
| Disallowed (%) | - | - | 0.05 | 0.05 | - | 0.01 | 0.02 |

§Resolution used for final atomic model refinement; model building and interpretation were carried out using local map resolutions. ◊No map sharpening was performed for model building or refinement. ¶RSCC, real-space correlation coefficient.

OXA1L/MITRAC complex. Both structures resolve major contact sites of the insertase OXAL1 at the side-chain or secondary-structure level (Fig. 7a), similar to that seen in a previous mtRNC–OXA1L structure[4], while the biochemical data show the presence of both C12ORF62 and MITRAC12 (Fig. 6b), confirming the presence of major MITRAC components[2,14]. The closed and open states of the COX1–mtRNC–OXA1L/MITRAC complex differ by a large-scale rotation between the mtRNC and OXA1L/MITRAC (Fig. 6d and Extended Data Fig. 6f). In the cell, OXA1L/MITRAC is integrated in the IMM and the rotational movement would translate the associated mtRNC away from the membrane in the open state. As a consequence, the space between ribosome and membrane opens up, which may be important to, for example, accommodate different folding intermediates of the nascent chain or to regulate access of auxiliary factors to the nascent chain. Because of the low resolution in the membrane-integral region, however, we cannot define an absolute orientation of the mtRNCs relative to the membrane. In the reported mtRNC–OXA1L structure by Itoh et al.[4], the mtRNC adopts a half-closed orientation—halfway between the present open and closed states (Extended Data Fig. 6f, bottom), indicating that the extent of opening relative to the membrane may be incrementally fine-tuned to adapt to different translational states. In line with this idea, cryo-electron tomography data showed that human mitochondrial ribosomes sample a large range of orientations versus the membrane in situ[31].

Sorting of the cryo-EM data strongly improved the density at the tunnel exit in the open state, resolving mtLSU proteins uL23m, uL24m and mL45m and the nascent chain in this area at 4–6-Å local resolution and allowing us to trace the COX1 nascent chain complex to about residue 37 from the PTC (Figs. 6d and 7a–c,f and Extended Data Figs. 6e and 7b). In contrast, the exit area appears highly dynamic in the closed state, as several proteins of the mtLSU are only partially resolved here and the nascent chain could be only followed to residue 20 (Figs. 6d and 7f and Extended Data Figs. 6e and 7a). The closed state accounts for a large fraction of OXA1L/MITRAC-bound mtRNC particles (~50%; the open state accounts for only ~15%), while the biochemical and profiling data showed that the majority of COX1 nascent chain intermediates are substantially longer than 20 aa (Fig. 4)[2]. Therefore, the closed state most likely comprises a mixture of nascent chain lengths extending well beyond the visible 20 residues.

In addition to the global differences in mtRNC membrane arrangement, the present COX1–mtRNC–OXA1L/MITRAC structures also show local differences in the key functional tunnel exit region from the reported structure of the mtRNC–OXAL1 complex, which was obtained by nonselective purification of actinonin-stalled mtRNCs with unspecified nascent chains[4]. In the mtRNC–OXAL1 structure, the N-terminal tail (NTT, residues ~37–51) of mL45 was found to insert into the lower part of the tunnel and form a constriction site there. Here and in the closed state, we observe only very weak density for this tail, which most likely reflects flexibility of the tail and indicates that this mitochondria-specific constriction site formed by the mL45 NTT may be dynamic (Extended Data Fig. 7d). Importantly, we were able to trace the linker of uL23m connecting the mtLSU-bound N-terminal domain with its C-terminal α-helix in contact with mL45 in the open COX1–mtRNC, showing that uL23m together with uL24m and mL45 forms a tight vestibule at the tunnel exit in the open state, which tightly wraps around the nascent chain and may slow translation (Fig. 7b–e,f). Notably, the interior of this tight vestibule is amphipathic, with mL45 contributing negatively charged residues while uL23m and uL24m largely provide hydrophobic patches. In the mtRNC–OXAL1 structure, the uL23m linker is unresolved and mL45 adopts a different position away from the nascent chain, overall resulting in a more open 'loose' vestibule, which may be better compatible with rapid translation (Fig. 7f). Both the uL23m linker and mL45 are unresolved in the vestibule region of the closed COX1–mtRNC, suggesting a highly flexible loose vestibule (Fig. 7f). In contrast, the vestibule is shown to be highly structured and

blocked by the uL23m linker before the onset of translation, that is, in the absence of a nascent chain (Fig. 7f)[4,32].

Most strikingly, the present open COX1–mtRNC–OXA1L/MITRAC structure indicates cotranslational folding of the nascent chain into a small α-helix in the vestibule (Fig. 7b,c). The observation of an α-helical element suggests that the open structure with its tight vestibule represents a stable translation intermediate reflecting translational pausing or slowing, potentially stabilized by interactions between nascent chain and the vestibule. The nascent chain density at ~6-Å local resolution does not allow identifying the amino acid sequence of the α-helix but the hydrophobic and negatively charged interior of the vestibule suggests that the enclosed nascent chain comprises a short sequence with complementary amphipathic properties, that is, combining hydrophobic and positively charged residues (Fig. 7d,e). For COX1, sequences with such properties correspond to the short N-terminal α-helix (COX1 α0, residues 2–7) and the N termini of matrix-facing TMH1, TMH3 and TMH9 (Fig. 7e and Extended Data Fig. 7e). Remarkably, emergence of COX1 α0 in the vestibule correlates with a translational pausing or slowing event according to the ribosome profiling data (codons 26–36 from the start codon; Fig. 7g). Translational pausing at this early point could provide a time window for mtRNC binding to OXA1L and associated assembly factors at the IMM, which is thought to occur after the onset of translation[4,32]. For COX3, the ribosome profiling data similarly indicate early slowing (codons 27–35) and major translational pausing (codons 35–53), correlating with placement of amphipathic and positively charged motifs at the N terminus of the nascent chain (COX3 residues 1–13) in the vestibule. Here, the early pausing might additionally facilitate the removal of the N-terminal formylmethionine, as COX3 is the only mitochondrial-encoded protein shown to lack the formyl group or the entire formylmethionine[33]. In the case of COX2, the major pausing event at codons 80–100 coincides with location of similarly amphipathic amino acids (around residue 45) in the vestibule. Overall, this suggests that the amphipathic nascent chain motif with positive charge may have a role in pausing events throughout mitochondrial translation elongation.

## Discussion

We combined cryo-EM and ribosome profiling to study mitochondrial translation dynamics during membrane insertion. Our MNase-based profiling (adapted from a previous study[19]) yielded unbiased mRNA-decoding snapshots, improved for transient stalling by MNase versus RNase1 and validated by antibiotic and snap-freezing comparisons, modified genome alignment and [$^{35}$S] labeling resemblance (Fig. 2a,b)[34–39]. In vitro silencing and intermediates further confirmed the findings. Unlike cytosolic data, the mitochondrial profiles show minimal start and stop accumulations because of leaderless mRNAs, LRPPRC positioning and rapid recycling[1,23,24,26,28,35,36,38,39]. We found that ribosomes accumulate similarly across mRNAs, especially at TMH release from the exit tunnel (Fig. 4), correlating read densities with COX1 and COX2 fragment sizes and indicating slowed translation upon TMH emergence.

TMH-related changes in translation occur natively and have been reported for both prokaryotic and eukaryotic translation systems[40–42]. Our data now show TMH synthesis-related changes in translation speed for mitochondrial mRNAs of each OXPHOS complex. This was particularly apparent for the first TMHs and differed in complexity depending on the N-terminal topology. Our profiling results suggest that retaining the N terminus in the matrix decreases translation speed at the level of emergence of the peptide from the exit tunnel (Fig. 5c,d). TMHs are passed across the lipid phase as single units (when N and C termini are facing the IMS) or pairwise. This process is supported by the topological region connecting the TMHs, which must be translocated to the IMS[40,41,43]. This topology is apparent at the level of translation speed with reoccurring patterns spaced by approximately 50 codons correlating with the average size of two TMHs spaced by an

interhelical loop (Fig. 5c,d). This reduction in translation speed may allow for recruitment of necessary nuclear-encoded protein factors assisting in membrane insertion and biogenesis of OXPHOS subunits or factor-independent diffusion of TMHs into the lipid bilayer[40,43].

Focusing on the cotranslational insertion of COX1, the profiling approach and the structural analysis allow for a mechanistic model for the coupling of translation states and mtRNC dynamics during membrane insertion of COX1 (Fig. 7g). At the onset of translation (Fig. 7g, stage I), our profiling data identified a major COX1 translation pause at codons ~0–10, coinciding with the previously reported blockage of the mitochondrial ribosomal exit tunnel by protein mL45 (refs. 44,32). At this stage, mL45 forms a short α-helix interacting specifically with the tunnel's primary constriction site, similar to that seen for pausing sequences in bacterial and cytosolic translation, indicating that mL45 contributes to the initial pausing[45,46]. Cryo-electron tomography data showed that only a minor mitoribosome fraction in human mitochondria is not associated with the membrane, likely representing initiation complexes[31]. Supporting this interpretation, cryo-EM structures show that, in initiating ribosomes, mL45 adopts a conformation unfavorable for OXA1L binding, whereas translation of a few amino acids induces a conformation switch in mL45 to a state favorable for binding, while still blocking the tunnel (Extended Data Fig. 6f)[44,32]. Consequently, this initial pausing might serve to prevent premature exposure of the hydrophobic nascent chain during membrane targeting, analogous to mechanisms described for bacterial and cytosolic ribosomes, which use bacterial-specific mRNA features and the eukaryotic signal recognition particle system to slow and pause translation, respectively[47,48].

A second major pause occurs around codons ~25–35, precisely when the N-terminal helix of COX1 (COX1 α0) enters and folds within the mitoribosomal vestibule, as likely observed in our open COX1–mtRNC–OXA1L/MITRAC structure (Fig. 7b,c,g, stage II). Nascent chain entry induces vestibule opening, tightly accommodating COX1 α0, which comprises an amphipathic motif featuring hydrophobic and positively charged patches complementary to the vestibule. These complementary interactions, together with other features such as mRNA secondary structure, may contribute to this and later pausing events[28]. Actually, our open COX1–mtRNC–OXA1L/MITRAC structure may present an averaged state of several pausing events, where a helix with an N-terminal amphipathic motif enters the vestibule, as observed in our profiling data for TMH3 and TMH9 of COX1 and for TMHs of other OXPHOS subunits such as COX2 and COX3. The early pausing around codons ~25–35 occurring while the nascent chain is still protected, yet primed to exit the vestibule, may ensure completion of targeting to OXAL1 and associated assembly factors in the membrane. Importantly, local conformational changes in the vestibule upon nascent chain folding correlate with a global open mtRNC configuration, causing the COX1–mtRNC to tilt away from membrane-bound OXA1L/MITRAC and creating an expanded gap that may be important to facilitate access of auxiliary factors from the mitochondrial matrix.

Conversely, extended regions in our ribosome profiling data with lower ribosome occupancy suggest periods of efficient translation, which are likely reflected by the closed COX1–mtRNC–OXA1L/MITRAC structure (Fig. 7g, stage III). In this state, the vestibule adopts a more relaxed configuration and the nascent chain remains unresolved beyond the exit tunnel, indicative of increased local dynamics. These local changes coincide with global closure, where the mtRNC tilts toward MITRAC and the membrane, thereby more shielding the nascent chain, which may serve to impede unwanted matrix interactions and promote efficient translation. A similar closed configuration is observed in bacterial RNC structures with the homologous YidC insertase by Kedrov et al.[49]. Itoh et al.[4] observed a slightly more open, half-closed configuration in their unspecified mtRNC–OXA1L structure, while the bacterial SecYEG translocon even formed a continuous conduit with the ribosomal exit tunnel, fully isolating the nascent chain during insertion[50–52].

In conclusion, our cryo-EM analysis provides structures of native nascent-chain-specific mtRNCs translating COX1 in complex with OXA1L/MITRAC. The improved biochemical homogeneity enabled us to resolve cotranslational folding of COX1 within an open mtRNC–OXA1L/MITRAC state, revealing that nascent chains can form α-helices in the extended vestibule of the mammalian mitoribosome. Importantly, this cotranslational folding appears tightly linked to local and global functional dynamics of the mtRNC. Local closure of the vestibule around the folded nascent chain correlates with major pausing events identified by our profiling data, whereas the mtRNC–OXA1L/MITRAC complex adopts an open overall arrangement that facilitates access to the nascent chain. Our data also indicate that the positioning of helices containing N-terminal amphipathic motifs within the vestibule—observed in the open COX1–mtRNC structure—correlates with several pausing events during the translation of COX1 and other OXPHOS subunits. However, the precise role of these amphipathic motifs in regulating translation remains to be determined. Moreover, our structural findings indicate that dynamic nascent chains, indicative of efficient translation, correlate with a more relaxed vestibule and a closed overall arrangement of the mtRNC–OXA1L/MITRAC complex, effectively shielding the nascent chain from the matrix. Our biochemical analysis confirmed that the present structures include not only the general insertase OXA1L but also the major COX1-specific assembly factors C12ORF62 and MITRAC12. Future studies will be required to dissect the role of nascent chain interactions on folding and insertion and to resolve how the specific assembly factors contribute to the functional dynamics seen in the present COX1–mtRNC–OXA1L/MITRAC complexes. Our current approach of purifying nascent-chain-specific mtRNCs for structural analysis and monitoring translation by ribosome profiling represents a critical advance toward addressing these and other fundamental questions regarding cotranslational membrane insertion in human mitochondria.

## Online content

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

## Methods

### Human-derived cell line culture

HEK293-Flp-In T-Rex (HEK293T, Invitrogen, R70007) cells were cultured in DMEM, supplemented with 10% (v/v) FBS (Biochrom), 2 mM L-glutamine, 1 mM sodium pyruvate and 50 mg ml$^{-1}$ uridine and incubated at 37 °C with 5% $CO_2$. Cells were grown to density of maximal 90% confluency before usage for experiments.

### Protein electrophoresis and immunodetection

To determine the presence of ribosomal proteins and visualize translation activity and in vitro mRNA KD, SDS–PAGE was performed using either 4–12% NuPAGE Bis–Tris (Invitorgen) or 10–18% Tris–tricine gradient gel electrophoresis. Electrophoresis was followed by protein transferred to PVDF membranes (Merck Millipore) using a PEQLAB semidry blot system and exposure to a phosphor screen for digital autoradiography with the Typhoon imaging system (GE Healthcare) or antibody decoration.

### Isolation of mitochondria from HEK293T cells

Mitochondria for in vitro translation experiments were isolated as previously described[17]. In brief, HEK293 cells were detached in ice-cold PBS, pelleted, resuspended and washed in ice-cold isotonic buffer (10 mM MOPS pH 7.2, 225 mM sucrose, 75 mM mannitol and 1 mM EGTA), before determination of cell weight. Cell were resuspended at 1 g per 5 ml in cold hypotonic buffer (10 mM MOPS pH 7.2, 100 mM sucrose, 2 mM PMSF and 1 mM EGTA) and allowed to swell for 6 min on ice before cell were opened in a Dounce glass homogenizer. Homogenized cells were equilibrated with 1.1 ml of hypertonic buffer per 5 g of cells before doubling volume with isotonic buffer supplemented with 2 mM PMSF and 2 mg ml$^{-1}$ of BSA. Mitochondria were enriched in the supernatant by two differential centrifugation steps at 1,000$g$ for 10 min at 4 °C before sedimentation of mitochondria at 11,000$g$ for 10 min at 4 °C. Enriched mitochondrial pellets were washed twice in isotonic buffer. Protein concentration was determined by Bradford assay.

### In vitro protein import

First, 1 mg of mitochondria were resuspended at a concentration of 1 mg ml$^{-1}$ in import buffer (20 mM HEPES, 100 mM mannitol, 80 mM KCl, 5 mM MgCl$_2$, 10 mM sodium succinate, 10 mM malate, 5 mM ATP, 6 mM creatine phosphate, 0.625 mg ml$^{-1}$ creatine kinase, 1 mg ml$^{-1}$ BSA and 5 mM NADH, pH 7.4). Import was initiated by addition of 2 μM Jac1-MO hybrids. Control samples were incubated with HEPES–mannitol buffer. Reactions were incubated 30 min at 37 °C with 500 rpm shaking and terminated by differential centrifugation at 10,000$g$ for 10 min at 4 °C.

### In vitro [35S]methionine labeling of newly synthesized mitochondrial transcripts

Sedimented mitochondria from import reactions are resuspended at 1 mg ml$^{-1}$ in Translation buffer (25 mM HEPES, 100 mM mannitol, 80 mM KCl, 5 mM MgCl$_2$, 10 mM sodium succinate, 1 mM potassium phosphate, 5 mM ATP, 6 mM creatine phosphate, 0.625 mg ml$^{-1}$ creatine kinase, 1 mg ml$^{-1}$ BSA, 0.02 mM GTP, 0.15 mM amino acid mix (lacking methionine) and 100 μg ml$^{-1}$ emetine, pH 7.4). All samples were incubated 5 min 37 °C before the addition of 100 mCi ml$^{-1}$ [35S]methionine. Newly synthesized translation products were labeled with [35S]methionine for 1 h at 37 °C and translation was stopped by the addition of 100 μg ml$^{-1}$ CHL and 10 min of further incubation. Mitochondria were sedimented at 10,000$g$ for 10 min at 4 °C; supernatants were removed and pellets were flash-frozen in liquid nitrogen.

### In vivo [35S] methionine labeling of newly synthesized mitochondrial transcripts in HEK293T cells

HEK293 cells were seeded at 500,000 cells in T25 Flask (Thermo) and grown for 2 days. Before incubation with [35S]methionine, cells were incubated twice for 10 min in fetal calf serum (FCS)-free DMEM. For [35S] methionine labeling, media were exchanged with DMEM supplemented with dialyzed FCS lacking methionine and 100 μg ml$^{-1}$ of either emetine or cycloheximide. Translation negative controls were supplemented with 100 μg ml$^{-1}$ CHL. HEK cells were incubated for 10 min before the addition of 200 mCi ml$^{-1}$ of [35S]methionine. Translation products were labeled for 1 h at 37 °C with 5% $CO_2$. Cells were harvested by scraping and equal protein amounts were diluted in SDS loading buffer. Proteins were separated on 10–18% Tris–tricine gradient gels and western blot and autoradiography were performed as described above. Autoradiographic signal intensities were quantified using ImageQuantTL version 8.1 (GE Healthcare).

### Isolation of mitochondrial mRNA ribosome footprints from in vivo and in vitro samples

Wild-type HEK293T cells were cultured as described above. Then, 3 days before the experiment, HEK293T cells were seeded at 14 × 10$^6$ cells on a 15-cm cell culture dish. HEK293T cells were seeded in T25 cell culture flasks. On the day of the experiment, mitochondrial and cytosolic translation was stalled either by 10 min of incubation at 37 °C, 5% $CO_2$ in the presence of 100 μg ml$^{-1}$ cycloheximide and 100 μg ml$^{-1}$ CHL or the medium was removed by pouring and 15-cm cell culture dishes were frozen by floating on liquid nitrogen. Cells were isolated by scraping into 1 ml of isolation buffer containing 100 μg ml$^{-1}$ cycloheximide and 100 μg ml$^{-1}$ CHL. Solubilization of organelles was supported by three passages through a 26G needle. In vitro translation samples were directly thawed by the addition of 1 ml of lysis buffer and pipet resuspension. Large fragments and insoluble fractions were sedimented at 20,000$g$ for 10 min at 4 °C. For in vitro translation, samples of pelleted mitochondria were lysed to at a concentration of 5 mg ml$^{-1}$. For digestion, 1 ml of cleared lysate was incubated with 0.12 mg ml$^{-1}$ MNase for 1 h at 4 °C with overhead rotating. Digestion was stopped with the addition of 20 mM EGTA and monosomes were enriched by 2 h of ultracentrifugation at 35,000 rpm (SW41Ti) on a 5–40% sucrose gradient before isolating fractions. Then, 10 μl of each fraction was isolated and loaded on a NuPAGE 4–12% Bis–Tris (1.0 mm) midi protein gel. All fractions were flash-frozen in liquid nitrogen and stored at −80 °C.

### Library generation for sequencing of enriched mitochondrial ribosomal footprints

Library preparation was performed as described in Günnigmann et al.[21]. In short, mitochondrial monosome containing fractions were thawed on ice and incubated with one volume of preheated acid phenol (Ambion) and 1/11 volume of 20% SDS (Ambion) at 65 °C 5 min. Separation of the aqueous layer was achieved by 2 min of centrifugation at maximum speed. The aqueous layer was incubated with one volume of acid phenol (Ambion) for 5 min. Centrifugation was repeated and the aqueous layer was incubated with 0.5 volumes of chloroform, before mixing and incubating for 2 min room temperature. After layer separation by centrifugation, RNA was precipitated from the aqueous layer by incubation with 1/9 volume of 3 M sodium acetate pH 5.5, one volume of isopropanol and 2 μl of Glycoblue for 2–16 h at −80 °C. Nucleic acids were precipitated by centrifugation at 20,000$g$ for 1 h at 4 °C; pellets were aspirated and washed with ice-cold 70% RNAse-free ethanol. Dried pellets were then resuspended in 10 mM Tris pH 7. Nucleic acids were separated on either 10% or 15% TBE urea polyacrylamide gels (Invitrogen) and bands corresponding to mitochondrial ribosomal footprints were isolated by gel extraction.

### Gel extraction of ribosomal footprints

TBE urea polyacrylamide gels were stained with SYBR gold (Invitrogen). Nucleic acids were visualized on a blue-light screen and size-corresponding areas were excised with RNAse-free scalpels. Gel pieces were centrifuged at 20,000$g$ for 5 min at 4 °C in gel breaker tubes (Ist Engineering). Broken gel pieces were incubated in 10 mM

Tris pH 7 for 15 min at 70 °C (1,400 rpm). Supernatants were isolated using Corning Costar Spin-X columns and RNA was precipitated as described above.

## Combined dephosphorylation and linker ligation

Resuspended RNA footprints were dephosphorylated using T4 polynucleotide kinase (New England Biolabs) as described by the supplier. For linker ligation, dephosphorylated footprints were incubated with 50% PEG MW 8000, T4 RNA ligase buffer, murine RNAse inhibitor and 1 µM 3L1 adenylated linker for 2 h at 22 °C. Footprint isolation was performed as described above.

## Generation and circularization of single-stranded DNA (ssDNA) from isolated RNA footprints

Ligated footprints were incubated with dNTP mix, 12.5 µM linker RT, FSB buffer (Invitrogen), murine RNAse inhibitor (New England Biolabs), 0.01 M DTT and 2 µl of Superscript III (Invitrogen) for 30 min at 50 °C. The reaction was terminated by the addition of 1 N NaOH (Ambion). Precipitated ssDNA was resuspended in 10 mM Tris pH 8 and incubated together with CircLigase buffer (Biozym), 0.1 mM ATP, 5 mM MnCl$_2$ and 100 U CircLigase (Biozym) for 2 h at 60 °C. Circularization was quenched at 80 °C.

Circularized footprints were PCR-amplified before sequencing and barcoded by incubating each sample with 1 µM barcode primer, followed by PCR amplification for 9–12 cycles.

## Sequencing of generated ssDNA libraries

Concentration of isolated libraries were determined using bioanalyzer and a double-stranded DNA high-sensitivity Qubit assay according to the manufacturer's instructions. Libraries were pooled according to recommendations for multiplexing on Illumina NextSeq system and sequenced according to instructions offered by Illumina.

## Data analysis

Samples were demultiplexed and resulting FASTQ files of adaptor sequences were removed using cutadapt (version 3.2) and the following command: cutadapt --cores=11 -m15 --discard-untrimmed -O6 -a ATCG-TAGATCGGAAGAGCACACGTCTGAACTCCAGTCAC -o '<output_path>/'outfile.fastq.gz' '<input_path>/infile.fastq.gz' 1>'<output_path>/'Cutadapt_report.txt. Unique molecular identifiers were handled as described in Bertolini et al.[19] using the published scripts and commands. Noncoding RNAs were removed using bowtie2 and reference FASTA files containing sequences of noncoding RNAs were obtained using the following command: bowtie2 -p16 -t -x '<ref_file_path>' -q 'infile.fastq.gz' --un '<outfile_path>/outfile.fastq' -S /dev/null 2>'<outfile_path>/Bowtie2_report.txt'. Sequences not matching noncoding RNAs were further aligned to a modified genome of the human chromosome M (GRCh38.p14) primary assembly using STAR (version 2.7.1) and the following command: STAR --runThreadN 7 --genomeDir '<ref_file_path>' --readFilesIn '<infile_path>/infile.fastq' --outSAMmultNmax 3 --outFilterType BySJout --outFilterMismatchNmax 2 --alignIntronMin 5 --outFileNamePrefix '<outfile_path>' --outReadsUnmapped Fastx --outSAMtype BAM SortedByCoordinate --outSAMattributes All XS --quantMode GeneCounts --twopassMode Basic --limitBAMsortRAM 1185598524. The modified chromosome M includes open reading frames for noncoding regions of genes and additional short poly(A) tails at the end of each mRNA to mimic post-transcriptional mRNA maturation. Assignment of the ribosomal A-site, P-site and E-site was performed using the Julia scripts described in Bertolini et al.[19], along with the removal of PCR duplicates based on UMIs, using the following command: julia -p 8 <script_path>script.jl -g <annotation_file_pat>/annotation_file.gff3 -o <out-put_path>/<input_path>/'infile.bam' -u -c 1. Further analysis was performed as described in Günningmann et al.[21] using the RiboSeqTools package (https://github.com/ilia-kats/RiboSeqTools).

## Isolation of RNC complexes using C12ORF62$^{FLAG}$ for cryo-EM

Cells were harvested and mitochondria were isolated as described previously[2,53]. Isolated mitochondria were lysed at 5 mg ml$^{-1}$ in solubilization buffer (130 mM sucrose, 100 mM KCl, 10 mM MgCl$_2$, 50 mM HEPES pH 7.4, 0.5% PCC, 1× Protease inhibitor and 100 µg ml$^{-1}$ CHL) for 30 min at 4 °C (1,000 rpm). Insoluble fractions were sedimented at 20,000$g$ and supernatants were loaded on sucrose cushions (34% sucrose (w/v), 100 mM KCl, 10 mM MgCl$_2$, 50 mM HEPES pH 7.4, 0.5% PCC, 1× protease inhibitor and 100 µg ml$^{-1}$ CHL). RNCs were sedimented at 29,400 rpm for 15 h at 4 °C in an SW40ti rotor. Pellets were recovered and resuspended in solubilization buffer with 20 pipet strokes. RNC complexes were then isolated using anti-FLAG M2 affinity gel (Sigma-Aldrich) and incubation for 1 h. Unbound fractions were removed and resin was washed with wash buffer (100 mM KCl, 10 mM MgCl$_2$, 50 mM HEPES pH 7.4, 0.1% PCC, 1× protease inhibitor and 100 µg ml$^{-1}$ CHL). Elution was carried out by 30 min of incubation of the resin in wash buffer supplemented with FLAG peptide at 4 °C. Eluates were further processed for cryo-EM.

## Cryo-EM analysis

COX1–mtRNC–MITRAC complexes for cryo-EM analysis were prepared by affinity-purification using FLAG-tagged C12ORF62$^{FLAG}$ as bait as described above. For cryo-EM grid preparation, 4.5 µl of the COX1–mtRNC–MITRAC complexes were applied to glow-discharged holey carbon grids (R3.5/1 from Quantifoil company) prefloated with a 2-nm continuous carbon film. After 30 s of sample adsorption time, grids were blotted manually for 2 s with prewetted filter paper before the application of another 4.5 µl of sample and 30 s of adsorption, followed by 8 s of manual blotting and plunge-freezing using a custom-made device.

Cryo-EM data were acquired on a Titan Krios G1 microscope (Thermo Fisher Scientific) equipped with a Falcon III camera (Thermo Fisher Scientific) and a spherical aberration corrector of the CETCOR-type (CEOS) at 300-kV acceleration voltage, and a nominal magnification of ×59,000, corresponding to a final calibrated pixel size of 1.16 Å. In total, 54,447 cryo-EM video images of 30–40 frames were collected in integration mode using a total dose of 30–45 e$^-$ per Å$^2$ and a defocus range of 0.2–2 µm.

Cryo-EM image processing was generally performed in RELION (versions 4 and 5) (Extended Data Fig. 5a), if not stated otherwise[54,55]. Cryo-EM video images were averaged using MotionCor2, global contrast transfer function parameters were estimated with Gctf (version 1.0.6) and particles were selected with Gautomatch (version 0.56)[56,57]. Selected particles were extracted fourfold binned to a pixel size of 4.64 Å and classified for particle quality by two-dimensional (2D) classification and global three-dimensional (3D) classification without and with particle image alignment. Particle images were then reextracted at the final pixel size of 1.16 Å and 3D structures were reconstructed for subsequent per-particle defocus refinement and per-particle motion correction by Bayesian polishing[58,59]. Polished particles were sorted by focused 3D classification with signal subtraction (FCwSS) for the mtSSU, followed by FCwSS for tRNA positions (Extended Data Fig. 5a,b), yielding five classes showing similar nascent chain density in the tunnel but distinct tRNA states (P, AP, AP*, H1 and H2) that were refined to high resolution by nonuniform refinement in cryoSPARC version 4.5 (Extended Data Fig. 5c)[60]. To better resolve the scattered density for the OXA1L/MITRAC complex at the tunnel exit of the LSU, the five particle classes were combined and sorted by FCwSS, firstly for the presence and absence of OXA1L/MITRAC and secondly for its global orientation, revealing open and closed orientations of OXA1L/MITRAC versus the mtRNC (Extended Data Fig. 5a,b,f); trials to sort into more than two conformational classes did not yield additional well-defined classes, suggesting that the present COX1–mtRNC–OXA1L/MITRAC complex predominantly samples the open and closed states. Subsequently, the open and closed mtRNC–OXA1L/MITRAC particles were classified

by FCwSS on the vestibule area. In this way, we were able to resolve this region, including the nascent chain, at the secondary-structure level for the open but not for the closed mtRNC population, indicating substantial vestibule and nascent chain dynamics in the latter case. The final open COX1–mtRNC–OXA1L/MITRAC and closed COX1–mtRNC–OXA1L/MITRAC particle populations were refined to high resolution by nonuniform refinement in cryoSPARC (version 4.5) (Extended Data Fig. 5c,e)[60].

Atomic models were built and refined for the AP* and H1 tRNA states and the open and closed COX1–mtRNC–OXA1L/MITRAC structures, following the same pipeline, if not stated otherwise. PDB 7QI5 was used as a general starting model for mtRNCs[61]. The mRNA, tRNAs and nascent chains in our structures represent a mixture of species occurring during native COX1 translation and we arbitrarily chose mRNA and tRNAs from PDB 7QI5 for starting models. The COX1 nascent chains were manually traced as polyalanine in WinCoot (version 0.9.8.95)[62]; the small N-terminal helix of the nascent chain in the open COX1–mtRNC–OXA1L/MITRAC complex was manually built into the lowpass-filtered density map (to 6 Å) using secondary-structure restraints.

Initial rigid-body fitting of starting models was performed in ChimeraX (version 1.9)[63], followed by flexible fitting using all-atom real-space refinement in WinCoot (version 0.9.8.95)[62] with all-molecule self-restraints at a 5-Å distance cutoff. For the COX1–mtRNC–OXA1L/MITRAC structures, starting models of uL23m, umL45 and the CTH and CTE of OXA1L from PDB 6ZM5 (ref. 4) were adjusted and refined manually in WinCoot (version 0.9.8.95)[62] using density maps lowpass-filtered to local resolution, applying secondary-structure and local distance restraints. The derived atomic models were processed with phenix.ready_set to prepare restraints and final models were then obtained by real-space refinement in PHENIX (version 1.21)[64] over five cycles with 300 iterations each, using reference-structure, secondary-structure and metal-coordination restraints (Table 1).

### Statistics and reproducibility

No statistical method was used to predetermine sample size. No data were excluded from the analyses. The experiments were not randomized and the investigators were not blinded to allocation during experiments and outcome assessment. Biochemical experiments (sucrose gradients, in organello translation, immunoprecipitations and radiolabeling) were performed $n = 3$ times independently with similar results ($n$ specified in figure legends; representative images shown where indicated). Ribosome profiling was performed with $n = 3$ independent biological replicates from separate cell cultures (technical details described above; RPKM normalization; metagene profiles as mean ± s.e.m.). Standard practices were followed for cyo-EM data collection and processing (micrographs: >50,000; particle numbers per class specified in Table 1; resolutions using FSC = 0.143 cutoff; local resolution maps generated).

### Materials availability

This study did not generate new unique reagents.

### Reporting summary

Further information on research design is available in the Nature Portfolio Reporting Summary linked to this article.

### Data availability

Data and materials can be obtained from the corresponding authors upon request. Cryo-EM micrographs and particle images were deposited to the EM Public Image Archive under accession code EMPIAR-13386. The cryo-EM maps and associated coordinates of atomic models generated in this study were deposited to the EM Data Bank and Protein Data Bank under the following accession codes: EMD-55836 (P), EMD-55837 (AP), EMD-54637/PDB 9S7B (AP*), EMD-54638/PDB 9S7C (H1), EMD-55838 (H2), EMD-54639/PDB 9S7D (open COX1–mtRNC–OXA1L/MITRAC) and EMD-54640/PDB 9S7E (closed COX1–mtRNC–OXA1L/MITRAC). Data from the sequencing analyses and all data underlying the sequencing-related figures were deposited to the Gene Expression Omnibus under accession code GSE303205. Source data are provided with this paper.

### Code availability

This paper does not report original code.

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

### Acknowledgements

We are grateful to our colleagues for their support and discussion. This study was funded by the European Research Council Advanced Grants MiXpress (101054637) to P.R., the Deutsche Forschungsgemeinschaft (DFG) SFB1565 (project number 469281184) to P.R. and the Max Planck Society (to P.R. and N.F.), supported by the DFG under Germany's Excellence Strategy (EXC 2067/1-390729940). G.K. acknowledges funding by the DFG (SPP2453, KR 3593/6-1, project 541620165). We thank H. Stark for providing access to infrastructure in his department at the Max Planck Institute for Multidisciplinary Sciences (MPI-NAT), as well as M. Lüttich and T. Koske (MPI-NAT, Department Stark) for their support with high-performance computing (HPC). We further acknowledge the Department of Molecular Biology (director: P. Cramer; MPI-NAT) for providing additional HPC resources and assistance.

### Author contributions

T.S., N.F. and P.R. conceptualized the study. T.S., V.P., I.K., S.D., N.F., L.D.C.-Z., A.S., N.N., O.U. and D.D. performed the experiments. T.S., I.K., V.P. and N.F. analyzed the datasets and prepared the figures.

T.S., N.F. and P.R. wrote the original draft. T.S., I.K., P.V., N.F. and P.R. reviewed and edited the final draft of the manuscript. N.F., G.K. and P.R. provided supervision.

## Funding

## Competing interests

The authors declare no competing interests.

## Additional information

**Extended data** is available for this paper at https://doi.org/10.1038/s41594-026-01803-w.

**Correspondence and requests for materials** should be addressed to Niels Fischer or Peter Rehling.

**a**

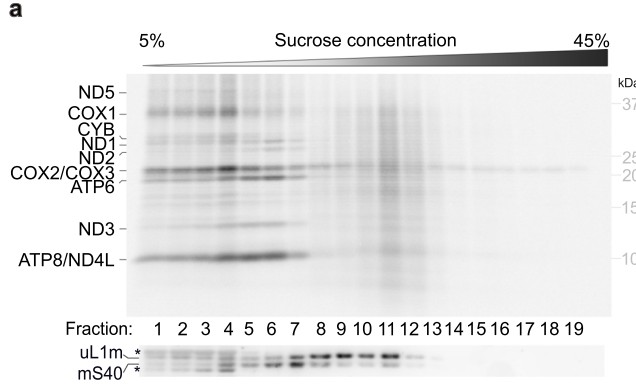

**b**

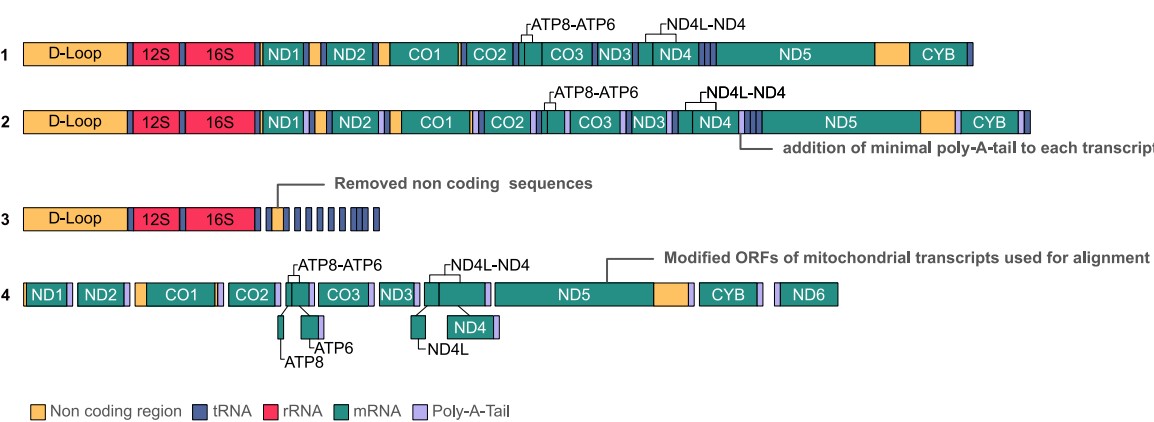

**c**

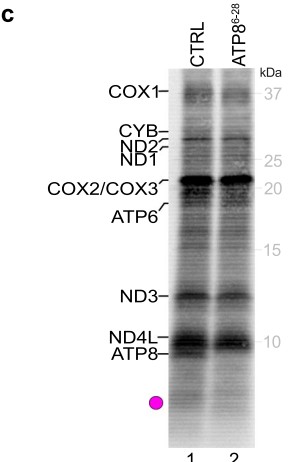

**Extended Data Fig. 1 | Supportive data for Profiling approach. a** Isolated mitochondria were subjected to [³⁵S]methionine labelling (without CHL treatment), solubilized, and MNase digested. Proteins were separated by sucrose density gradient centrifugation, fractionated, and analyzed by SDS-PAGE followed by digital autoradiography. Presence of ribosomal subunits was confirmed by western blotting with indicated antibodies (n = 3). **b** Schematic representation of the bioinformatical modifications to the mitochondrial genome carried out for footprint alignment. **c** [35S]methionine labelling was performed in purified mitochondria in the presence or absence of indicated morpholino chimera targeting ATP8 (pink circles) Samples were analyzed by SDS-PAGE followed by digital autoradiography (n = 3).

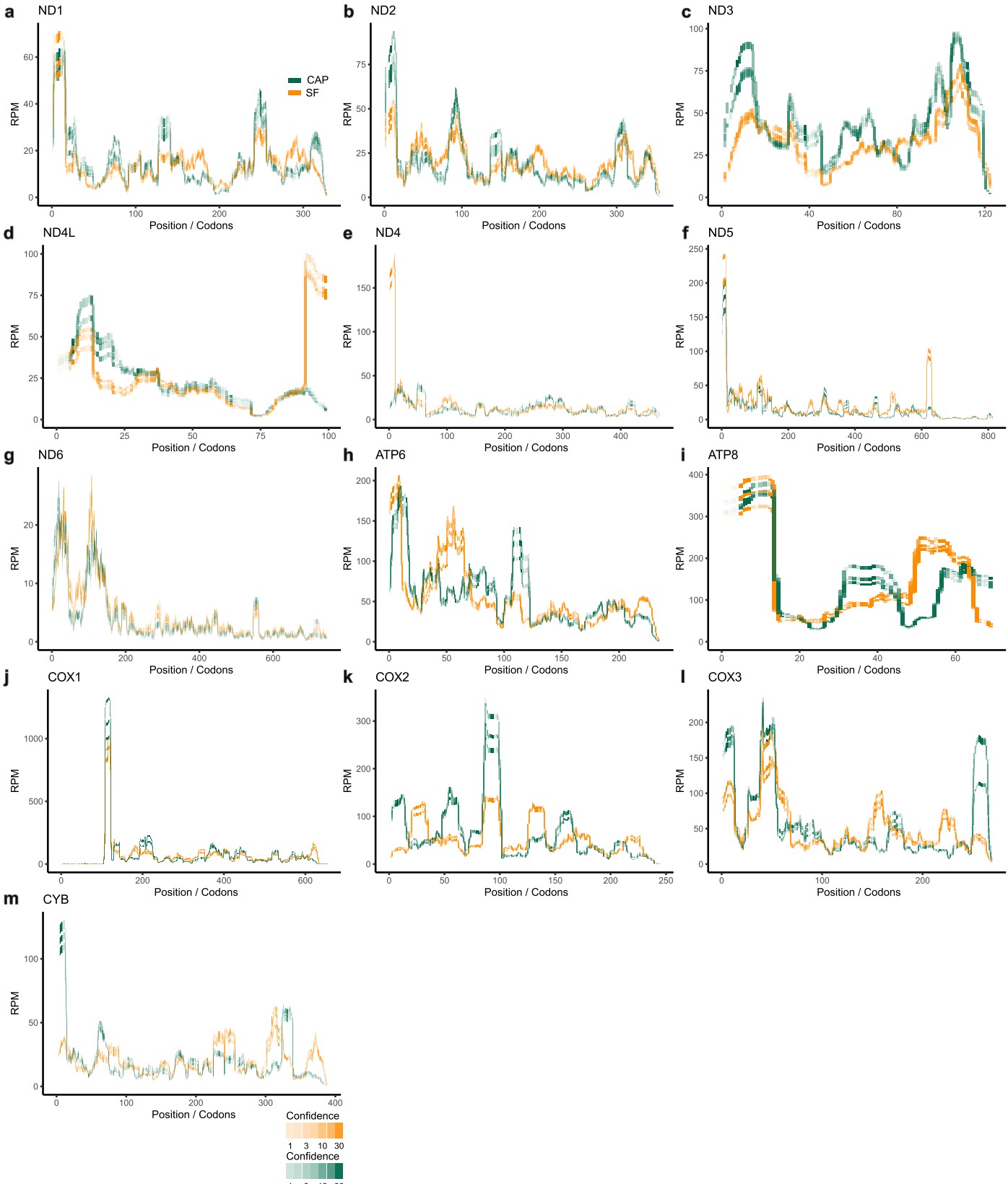

**Extended Data Fig. 2 | Alignment of Footprints isolated from mitochondrial mRNAs. a - m** Mitochondrial RNA footprints aligned to the 13 different mitochondrial mRNAs. For both CHL (green) and snap frozen samples lacking CHL (yellow) (n = 3).

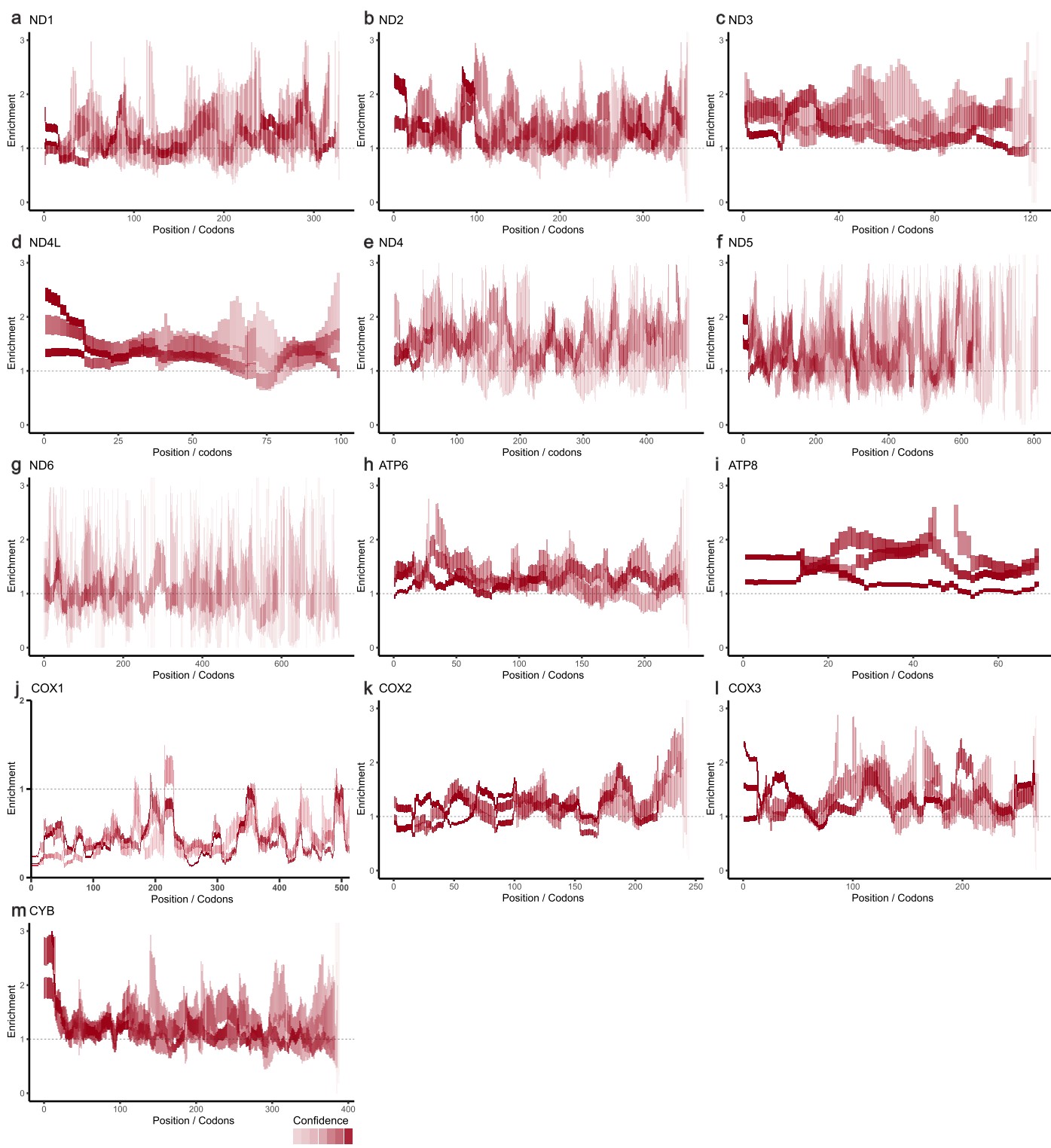

**Extended Data Fig. 3 | mRNAs enrichment profiles after COX1 silencing in purified mitochondria. a-m** Enrichment profiles of reads isolated form mitochondria treated with COX1[1-19] over the control condition, (n = 3).

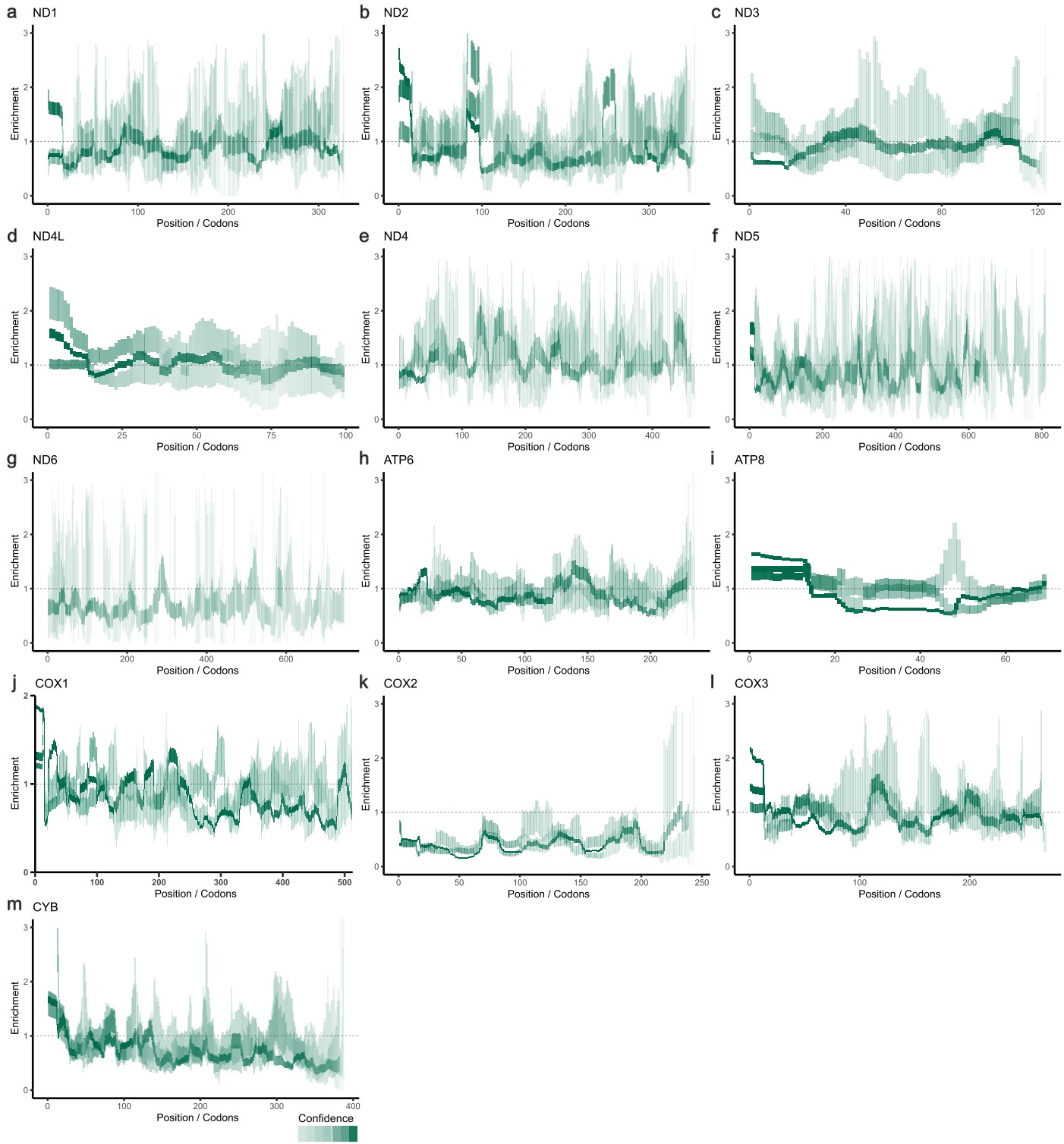

**Extended Data Fig. 4 | Enrichment profiles of mitochondrial mRNAs after COX2[1-23] silencing. a-m** Enrichment profiles of reads isolated form mitochondria treated with COX2[1-23] over the control condition, (n = 3). See also Extended Data Fig. 3.

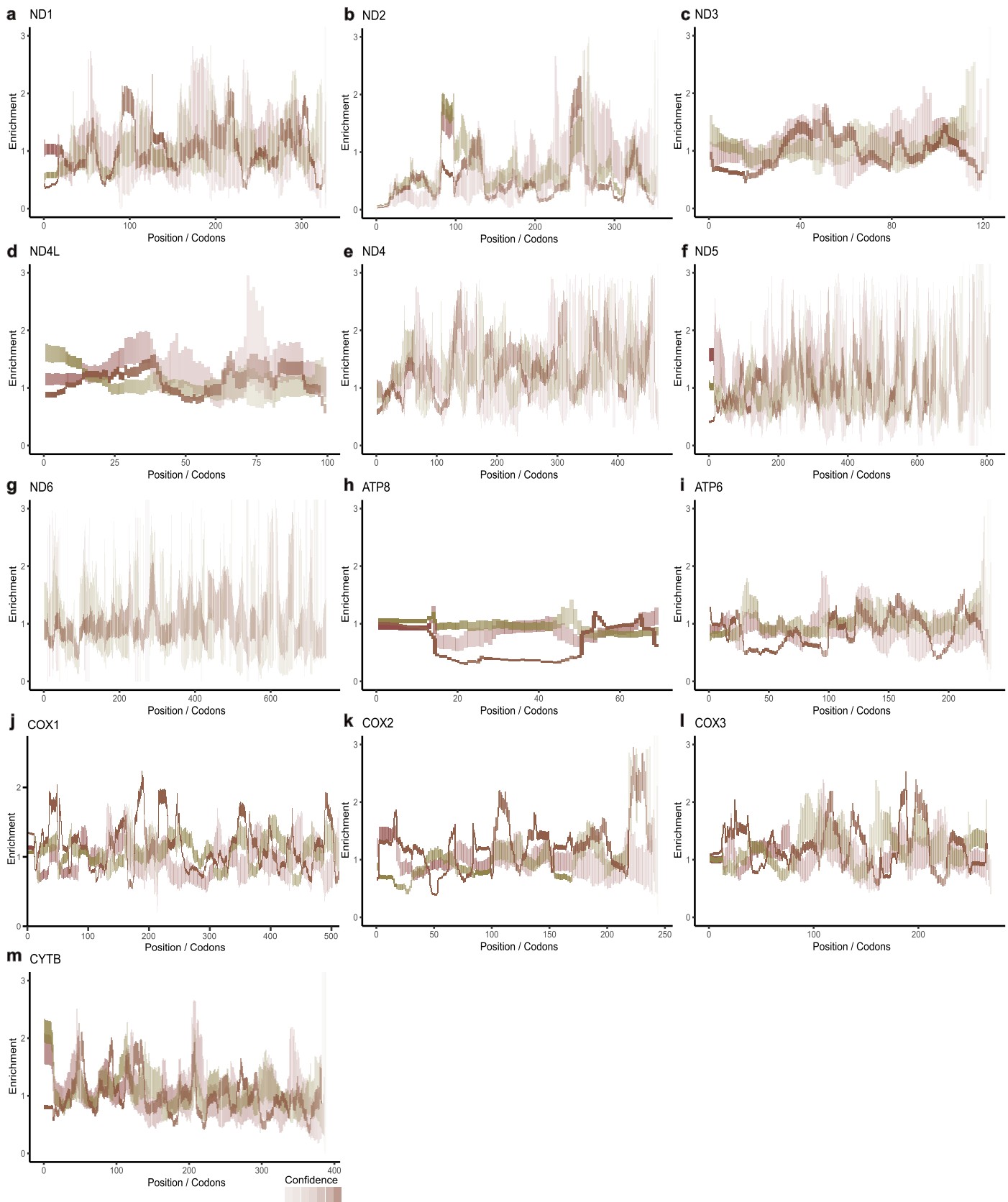

**Extended Data Fig. 5 | Enrichment profiles of mitochondrial mRNAs after ND2[12-29] silencing. a-m** Enrichment profiles of reads isolated form mitochondria treated with ND2[12-29] over the control condition, (n = 3). See also Extended Data Fig. 3.

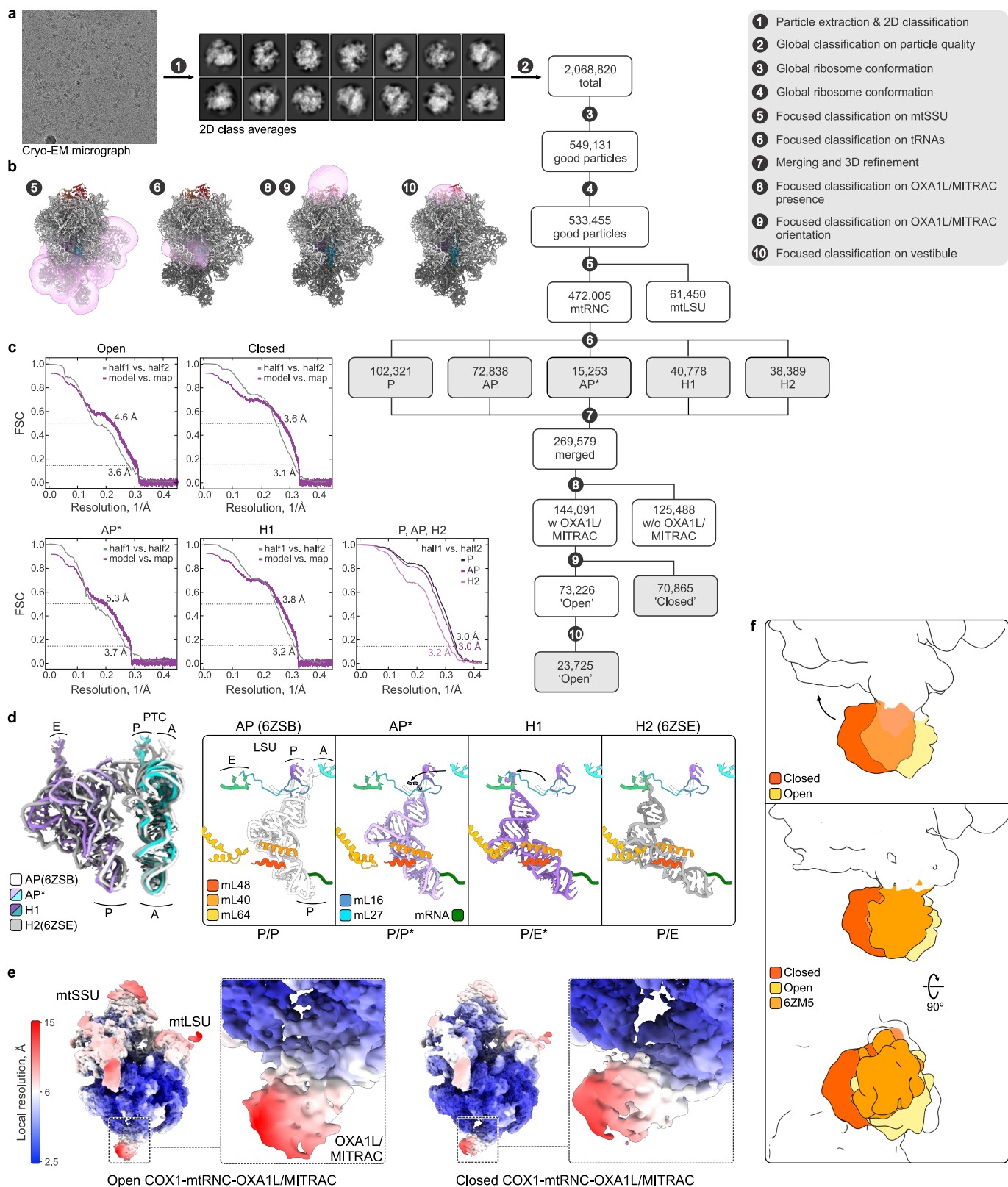

**Extended Data Fig. 6 | See next page for caption.**

**Extended Data Fig. 6 | Cryo-EM analysis of COX1-mtRNC-OXA1L/MITRAC complexes. a** Cryo-EM workflow. For details see Methods. **b** Masks used for focused classification. **c** Resolution estimation by Fourier-shell correlation (FSC). FSC curves are shown for the final cryo-EM reconstructions between half-maps ('h1 vs. h2') and between full maps and atomic models ('model vs. map'), respectively. Atomic models were built for novel states, that is AP*, H1, and the open and closed COX1-mtRNC-OXA1L/MITRAC structures ('Open' and 'Closed'). **d** Extended trajectory of tRNA movement through the mitochondrial ribosome. Left: Overall trajectory. Note the A-site tRNA, for which only the CCA-end moves slightly from the mtLSU's A- to P-loop upon peptidyl-transfer. Right: Extensive movement of P-site tRNA with both CCA-end and elbow region. A, P, E, tRNA binding sites. P*, E*, intermediate tRNA binding sites on mtLSU. PTC, peptidyl transferase center of the mtLSU. Models for classical pre-translocation state ('AP') and final hybrid state ('H2') are taken from PDBs 6ZSB and 6ZSE, respectively[29]. **e** Local resolution of COX1-mtRNC-OXA1L/MITRAC reconstructions. Local resolutions are plotted onto the locally filtered maps. **f** Global dynamics of COX1-mtRNC-OXA1L/MITRAC and mtRNC-OXA1L complexes. Top: Superposition of open and closed COX1-mtRNC-OXA1L/MITRAC by mtLSU illustrating the large-scale reorientation of OXA1L/MITRAC with respect to the mtLSU. Center & bottom: Superposition of open and closed COX1-mtRNC-OXA1L/MITRAC versus the reported mtRNC-OXA1L structure (EMD-112784)[4], aligned by mtLSU.

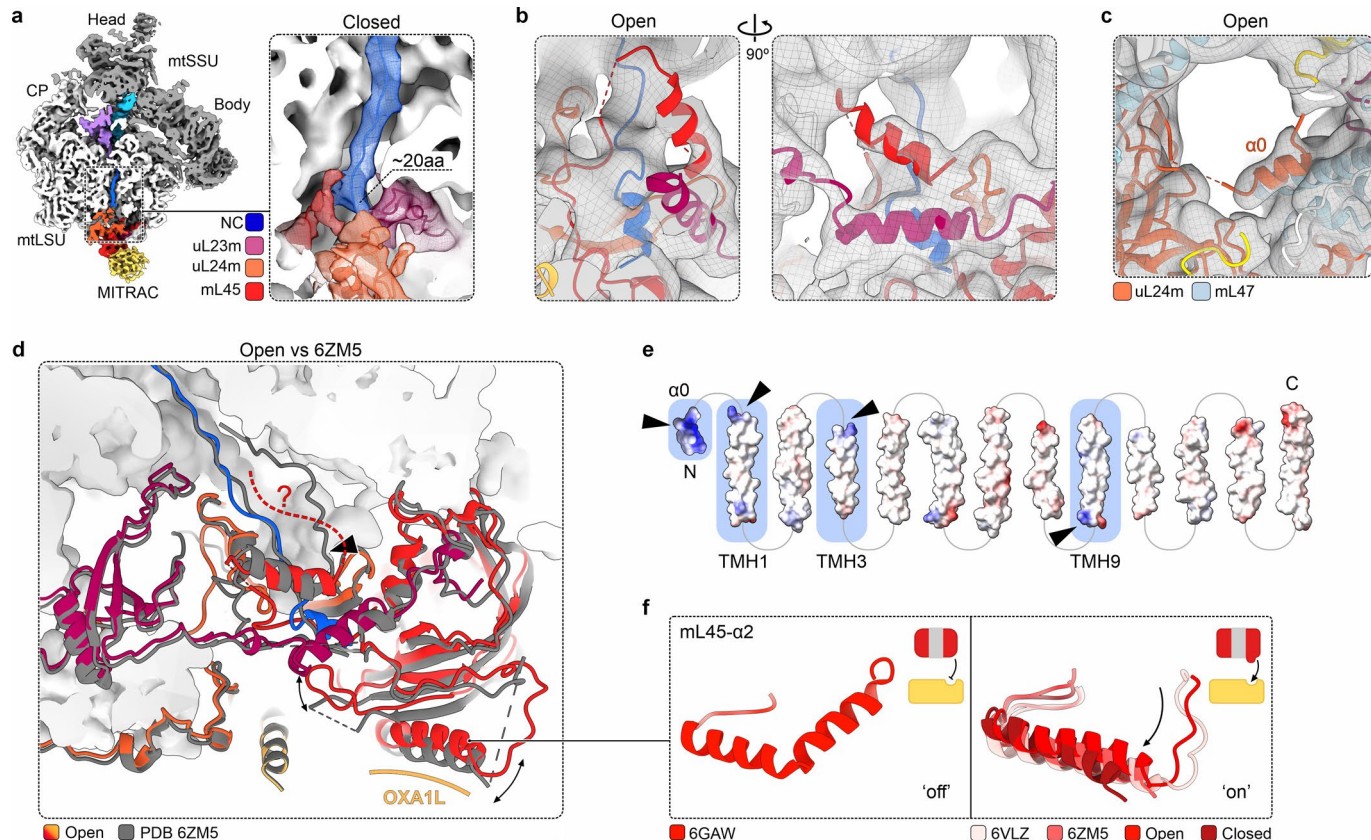

**Extended Data Fig. 7 | Details of vestibule, COX1 properties and dynamics of mL45. a** Nascent chain path in the closed COX1-mtRNC-MITRAC. The nascent chain can be traced from the PTC of the mtLSU to about residue 20. Semi-transparent mesh (in panels **A**, **B** and **C**), low-pass filtered cryo-EM densities (to 6 Å). **b** Close-up of vestibule area in the open COX1-mtRNC-MITRAC. Left: Nascent chain. Right: Vestibule. Densities are rendered at an intermediate threshold of 2.5 σ. **c** Location of uL24m N-terminal helix (α0). Local sorting for the vestibule enabled tracing the uL24m-α0, which interacts with mL47. **d** Comparison of vestibule in open COX1-mtRNC-MITRAC versus the reported half-closed mtRNC-OXA1L structure (PDB 6ZM5)[4]. '?', N-terminal tail of mL45, flexible in present COX1-mtRNC, while lining and constricting the tunnel in the previous unspecified mtRNC structure (arrowhead). Arrows,

major rearrangements of mL45 contributing to vestibule closure and OXA1L reorientation, respectively. **e** Amphipathic properties of COX1. Blue shades, N-terminal helix (α0) and transmembrane helices (TMHs) 1, 3 and 9, comprising a motif of positively charged and hydrophobic residues at their N-termini (arrow-heads). **f** Conformation of OXA1L-binding site on mL45 - helix 2 (α2) in cryo-EM structures of the initiating mitochondrial ribosome (left, PDB 6GAW Kummer et al.[32]), very early elongating mtRNCs without OXA1L (right, PDB 6VLZ Koripella et al.[44]), and later mtRNCs with OXA1L (right, PDB 6ZM5 Itoh et al.[4]) or OXA1L/MITRAC bound (right, present 'open' and 'closed' COX1-mtRNCs). Note the increased binding interface observed in all mtRNCs, indicating favorable OXA1L-binding properties ('on'), as compared to initiating ribosomes ('off').

# Reporting Summary

## Statistics

For all statistical analyses, confirm that the following items are present in the figure legend, table legend, main text, or Methods section.

| n/a | Confirmed | |
|---|---|---|
| ☐ | ☒ | The exact sample size (*n*) for each experimental group/condition, given as a discrete number and unit of measurement |
| ☐ | ☒ | A statement on whether measurements were taken from distinct samples or whether the same sample was measured repeatedly |
| ☒ | ☐ | The statistical test(s) used AND whether they are one- or two-sided<br>*Only common tests should be described solely by name; describe more complex techniques in the Methods section.* |
| ☒ | ☐ | A description of all covariates tested |
| ☒ | ☐ | A description of any assumptions or corrections, such as tests of normality and adjustment for multiple comparisons |
| ☐ | ☒ | A full description of the statistical parameters including central tendency (e.g. means) or other basic estimates (e.g. regression coefficient) AND variation (e.g. standard deviation) or associated estimates of uncertainty (e.g. confidence intervals) |
| ☒ | ☐ | For null hypothesis testing, the test statistic (e.g. *F*, *t*, *r*) with confidence intervals, effect sizes, degrees of freedom and *P* value noted<br>*Give P values as exact values whenever suitable.* |
| ☒ | ☐ | For Bayesian analysis, information on the choice of priors and Markov chain Monte Carlo settings |
| ☒ | ☐ | For hierarchical and complex designs, identification of the appropriate level for tests and full reporting of outcomes |
| ☒ | ☐ | Estimates of effect sizes (e.g. Cohen's *d*, Pearson's *r*), indicating how they were calculated |

*Our web collection on statistics for biologists contains articles on many of the points above.*

## Software and code

Policy information about availability of computer code

| Data collection | Cryo-EM data collection was performed using software EPU 2.3 (ThermoFisher) and CETCORPLUS 4.6.9 (CEOS). All software is commercially available.Sequencing data was collected on ilumina Nextseq 550 |
|---|---|
| Data analysis | All software used for cryo-EM data analysis has been described in Methods and is publicly available free-of-charge: RELION 4.0, GCTF 1.0.6, GAUTOMATCH 0.56, CryoSPARC 4.4.0, Coot 0.9.8.1, PHENIX 1.16-3549, UCSF ChimeraX 1.4 and 1.7, Matplotlib 3.5.3, Python 3.8, Python package scikit-learn 1.1.3. The script for principal component analysis of tRNA motions is available on GitHub (https://github.com/MolecularMachines-in-Motion/tRNA_PCA). Data analysis of sequenced footprints was performed useing custom pipeline freely available at https://rdrr.io/github/ilia-kats/RiboSeqTools/ |

For manuscripts utilizing custom algorithms or software that are central to the research but not yet described in published literature, software must be made available to editors and reviewers. We strongly encourage code deposition in a community repository (e.g. GitHub). See the Nature Portfolio guidelines for submitting code & software for further information.

## Data

Policy information about [availability of data](link)

All manuscripts must include a [data availability statement](link). This statement should provide the following information, where applicable:

- Accession codes, unique identifiers, or web links for publicly available datasets
- A description of any restrictions on data availability
- For clinical datasets or third party data, please ensure that the statement adheres to our [policy](link)

*Provide your data availability statement here.*

## Research involving human participants, their data, or biological material

Policy information about studies with [human participants or human data](link). See also policy information about [sex, gender (identity/presentation), and sexual orientation](link) and [race, ethnicity and racism](link).

| | |
|---|---|
| Reporting on sex and gender | n/a |
| Reporting on race, ethnicity, or other socially relevant groupings | n/a |
| Population characteristics | n/a |
| Recruitment | n/a |
| Ethics oversight | n/a |

Note that full information on the approval of the study protocol must also be provided in the manuscript.

# Field-specific reporting

Please select the one below that is the best fit for your research. If you are not sure, read the appropriate sections before making your selection.

☒ Life sciences ☐ Behavioural & social sciences ☐ Ecological, evolutionary & environmental sciences

For a reference copy of the document with all sections, see [nature.com/documents/nr-reporting-summary-flat.pdf](http://nature.com/documents/nr-reporting-summary-flat.pdf)

# Life sciences study design

All studies must disclose on these points even when the disclosure is negative.

| | |
|---|---|
| Sample size | The present study is based on cryo-EM data and therefore, does not require statistical methods do determine sample size. Statistical information on sequencing analyses is provided in the manuscript. |
| Data exclusions | Cryo-EM micrographs of bad quality, i.e. with max. resolution >4.5Å, were excluded. Fottprint sizes after sequencing not matching the mitochodnrial monosome were excluded form analysis. |
| Replication | Cryo-EM and sequencing data sets from three independent biological sample preparations were acquired and analyzed. |
| Randomization | Does not apply to our study, because we do not separate data into treatment groups. |
| Blinding | Does not apply to our study, because we do not group our samples. |

# Behavioural & social sciences study design

All studies must disclose on these points even when the disclosure is negative.

| | |
|---|---|
| Study description | n/a |
| Research sample | n/a |
| Sampling strategy | n/a |
| Data collection | n/a |

| Timing | n/a |
| Data exclusions | n/a |
| Non-participation | n/a |
| Randomization | n/a |

# Ecological, evolutionary & environmental sciences study design

All studies must disclose on these points even when the disclosure is negative.

| Study description | n/a |
| Research sample | n/a |
| Sampling strategy | n/a |
| Data collection | n/a |
| Timing and spatial scale | n/a |
| Data exclusions | n/a |
| Reproducibility | n/a |
| Randomization | n/a |
| Blinding | n/a |

Did the study involve field work? ☐ Yes ☒ No

## Field work, collection and transport

| Field conditions | n/a |
| Location | n/a |
| Access & import/export | n/a |
| Disturbance | n/a |

# Reporting for specific materials, systems and methods

We require information from authors about some types of materials, experimental systems and methods used in many studies. Here, indicate whether each material, system or method listed is relevant to your study. If you are not sure if a list item applies to your research, read the appropriate section before selecting a response.

## Materials & experimental systems

| n/a | Involved in the study |
|---|---|
| ☐ | ☒ Antibodies |
| ☐ | ☒ Eukaryotic cell lines |
| ☒ | ☐ Palaeontology and archaeology |
| ☒ | ☐ Animals and other organisms |
| ☒ | ☐ Clinical data |
| ☒ | ☐ Dual use research of concern |
| ☒ | ☐ Plants |

## Methods

| n/a | Involved in the study |
|---|---|
| ☒ | ☐ ChIP-seq |
| ☒ | ☐ Flow cytometry |
| ☒ | ☐ MRI-based neuroimaging |

## Antibodies

| | |
|---|---|
| Antibodies used | All antibodies used are indicated in the Materials and Methods section |
| Validation | All antibodies have been published previously |

## Eukaryotic cell lines

Policy information about cell lines and Sex and Gender in Research

| | |
|---|---|
| Cell line source(s) | HEK293 FITR cells were used for purification of monosomes used in downstream experiments for sequenciong and Cryo-EM |
| Authentication | commercial cell line |
| Mycoplasma contamination | No mycoplasma contaminations were detected. |
| Commonly misidentified lines (See ICLAC register) | n/a |

## Palaeontology and Archaeology

| | |
|---|---|
| Specimen provenance | *Provide provenance information for specimens and describe permits that were obtained for the work (including the name of the issuing authority, the date of issue, and any identifying information). Permits should encompass collection and, where applicable, export.* |
| Specimen deposition | *Indicate where the specimens have been deposited to permit free access by other researchers.* |
| Dating methods | *If new dates are provided, describe how they were obtained (e.g. collection, storage, sample pretreatment and measurement), where they were obtained (i.e. lab name), the calibration program and the protocol for quality assurance OR state that no new dates are provided.* |

☐ Tick this box to confirm that the raw and calibrated dates are available in the paper or in Supplementary Information.

| | |
|---|---|
| Ethics oversight | *Identify the organization(s) that approved or provided guidance on the study protocol, OR state that no ethical approval or guidance was required and explain why not.* |

Note that full information on the approval of the study protocol must also be provided in the manuscript.

## Animals and other research organisms

Policy information about studies involving animals; ARRIVE guidelines recommended for reporting animal research, and Sex and Gender in Research

| | |
|---|---|
| Laboratory animals | *For laboratory animals, report species, strain and age OR state that the study did not involve laboratory animals.* |
| Wild animals | *Provide details on animals observed in or captured in the field; report species and age where possible. Describe how animals were caught and transported and what happened to captive animals after the study (if killed, explain why and describe method; if released, say where and when) OR state that the study did not involve wild animals.* |
| Reporting on sex | *Indicate if findings apply to only one sex; describe whether sex was considered in study design, methods used for assigning sex. Provide data disaggregated for sex where this information has been collected in the source data as appropriate; provide overall numbers in this Reporting Summary. Please state if this information has not been collected. Report sex-based analyses where performed, justify reasons for lack of sex-based analysis.* |
| Field-collected samples | *For laboratory work with field-collected samples, describe all relevant parameters such as housing, maintenance, temperature, photoperiod and end-of-experiment protocol OR state that the study did not involve samples collected from the field.* |
| Ethics oversight | *Identify the organization(s) that approved or provided guidance on the study protocol, OR state that no ethical approval or guidance was required and explain why not.* |

Note that full information on the approval of the study protocol must also be provided in the manuscript.

# Clinical data

Policy information about clinical studies

All manuscripts should comply with the ICMJE guidelines for publication of clinical research and a completed CONSORT checklist must be included with all submissions.

| | |
|---|---|
| Clinical trial registration | *Provide the trial registration number from ClinicalTrials.gov or an equivalent agency.* |
| Study protocol | *Note where the full trial protocol can be accessed OR if not available, explain why.* |
| Data collection | *Describe the settings and locales of data collection, noting the time periods of recruitment and data collection.* |
| Outcomes | *Describe how you pre-defined primary and secondary outcome measures and how you assessed these measures.* |

# Dual use research of concern

Policy information about dual use research of concern

## Hazards

Could the accidental, deliberate or reckless misuse of agents or technologies generated in the work, or the application of information presented in the manuscript, pose a threat to:

No  Yes

☒ ☐ Public health

☒ ☐ National security

☒ ☐ Crops and/or livestock

☒ ☐ Ecosystems

☒ ☐ Any other significant area

## Experiments of concern

Does the work involve any of these experiments of concern:

No  Yes

☒ ☐ Demonstrate how to render a vaccine ineffective

☒ ☐ Confer resistance to therapeutically useful antibiotics or antiviral agents

☒ ☐ Enhance the virulence of a pathogen or render a nonpathogen virulent

☒ ☐ Increase transmissibility of a pathogen

☒ ☐ Alter the host range of a pathogen

☒ ☐ Enable evasion of diagnostic/detection modalities

☒ ☐ Enable the weaponization of a biological agent or toxin

☒ ☐ Any other potentially harmful combination of experiments and agents

# Plants

| | |
|---|---|
| Seed stocks | *Report on the source of all seed stocks or other plant material used. If applicable, state the seed stock centre and catalogue number. If plant specimens were collected from the field, describe the collection location, date and sampling procedures.* |
| Novel plant genotypes | *Describe the methods by which all novel plant genotypes were produced. This includes those generated by transgenic approaches, gene editing, chemical/radiation-based mutagenesis and hybridization. For transgenic lines, describe the transformation method, the number of independent lines analyzed and the generation upon which experiments were performed. For gene-edited lines, describe the editor used, the endogenous sequence targeted for editing, the targeting guide RNA sequence (if applicable) and how the editor was applied.* |
| Authentication | *Describe any authentication procedures for each seed stock used or novel genotype generated. Describe any experiments used to assess the effect of a mutation and, where applicable, how potential secondary effects (e.g. second site T-DNA insertions, mosiacism, off-target gene editing) were examined.* |

# ChIP-seq

## Data deposition

☐ Confirm that both raw and final processed data have been deposited in a public database such as GEO.

☐ Confirm that you have deposited or provided access to graph files (e.g. BED files) for the called peaks.

**Data access links**
*May remain private before publication.*
> For "Initial submission" or "Revised version" documents, provide reviewer access links. For your "Final submission" document, provide a link to the deposited data.

**Files in database submission**
> Provide a list of all files available in the database submission.

**Genome browser session**
(e.g. UCSC)
> Provide a link to an anonymized genome browser session for "Initial submission" and "Revised version" documents only, to enable peer review. Write "no longer applicable" for "Final submission" documents.

## Methodology

**Replicates**
> Describe the experimental replicates, specifying number, type and replicate agreement.

**Sequencing depth**
> Describe the sequencing depth for each experiment, providing the total number of reads, uniquely mapped reads, length of reads and whether they were paired- or single-end.

**Antibodies**
> Describe the antibodies used for the ChIP-seq experiments; as applicable, provide supplier name, catalog number, clone name, and lot number.

**Peak calling parameters**
> Specify the command line program and parameters used for read mapping and peak calling, including the ChIP, control and index files used.

**Data quality**
> Describe the methods used to ensure data quality in full detail, including how many peaks are at FDR 5% and above 5-fold enrichment.

**Software**
> Describe the software used to collect and analyze the ChIP-seq data. For custom code that has been deposited into a community repository, provide accession details.

# Flow Cytometry

## Plots

Confirm that:

☐ The axis labels state the marker and fluorochrome used (e.g. CD4-FITC).

☐ The axis scales are clearly visible. Include numbers along axes only for bottom left plot of group (a 'group' is an analysis of identical markers).

☐ All plots are contour plots with outliers or pseudocolor plots.

☐ A numerical value for number of cells or percentage (with statistics) is provided.

## Methodology

**Sample preparation**
> Describe the sample preparation, detailing the biological source of the cells and any tissue processing steps used.

**Instrument**
> Identify the instrument used for data collection, specifying make and model number.

**Software**
> Describe the software used to collect and analyze the flow cytometry data. For custom code that has been deposited into a community repository, provide accession details.

**Cell population abundance**
> Describe the abundance of the relevant cell populations within post-sort fractions, providing details on the purity of the samples and how it was determined.

**Gating strategy**
> Describe the gating strategy used for all relevant experiments, specifying the preliminary FSC/SSC gates of the starting cell population, indicating where boundaries between "positive" and "negative" staining cell populations are defined.

☐ Tick this box to confirm that a figure exemplifying the gating strategy is provided in the Supplementary Information.

# Magnetic resonance imaging

## Experimental design

**Design type**
> Indicate task or resting state; event-related or block design.

| Design specifications | *Specify the number of blocks, trials or experimental units per session and/or subject, and specify the length of each trial or block (if trials are blocked) and interval between trials.* |
|---|---|
| Behavioral performance measures | *State number and/or type of variables recorded (e.g. correct button press, response time) and what statistics were used to establish that the subjects were performing the task as expected (e.g. mean, range, and/or standard deviation across subjects).* |

## Acquisition

| Imaging type(s) | *Specify: functional, structural, diffusion, perfusion.* |
|---|---|
| Field strength | *Specify in Tesla* |
| Sequence & imaging parameters | *Specify the pulse sequence type (gradient echo, spin echo, etc.), imaging type (EPI, spiral, etc.), field of view, matrix size, slice thickness, orientation and TE/TR/flip angle.* |
| Area of acquisition | *State whether a whole brain scan was used OR define the area of acquisition, describing how the region was determined.* |

Diffusion MRI ☐ Used ☐ Not used

## Preprocessing

| Preprocessing software | *Provide detail on software version and revision number and on specific parameters (model/functions, brain extraction, segmentation, smoothing kernel size, etc.).* |
|---|---|
| Normalization | *If data were normalized/standardized, describe the approach(es): specify linear or non-linear and define image types used for transformation OR indicate that data were not normalized and explain rationale for lack of normalization.* |
| Normalization template | *Describe the template used for normalization/transformation, specifying subject space or group standardized space (e.g. original Talairach, MNI305, ICBM152) OR indicate that the data were not normalized.* |
| Noise and artifact removal | *Describe your procedure(s) for artifact and structured noise removal, specifying motion parameters, tissue signals and physiological signals (heart rate, respiration).* |
| Volume censoring | *Define your software and/or method and criteria for volume censoring, and state the extent of such censoring.* |

## Statistical modeling & inference

| Model type and settings | *Specify type (mass univariate, multivariate, RSA, predictive, etc.) and describe essential details of the model at the first and second levels (e.g. fixed, random or mixed effects; drift or auto-correlation).* |
|---|---|
| Effect(s) tested | *Define precise effect in terms of the task or stimulus conditions instead of psychological concepts and indicate whether ANOVA or factorial designs were used.* |

Specify type of analysis: ☐ Whole brain ☐ ROI-based ☐ Both

| Statistic type for inference<br>(See Eklund et al. 2016) | *Specify voxel-wise or cluster-wise and report all relevant parameters for cluster-wise methods.* |
|---|---|
| Correction | *Describe the type of correction and how it is obtained for multiple comparisons (e.g. FWE, FDR, permutation or Monte Carlo).* |

## Models & analysis

| n/a | Involved in the study |
|---|---|
| ☐ | ☐ Functional and/or effective connectivity |
| ☐ | ☐ Graph analysis |
| ☐ | ☐ Multivariate modeling or predictive analysis |

| Functional and/or effective connectivity | *Report the measures of dependence used and the model details (e.g. Pearson correlation, partial correlation, mutual information).* |
|---|---|
| Graph analysis | *Report the dependent variable and connectivity measure, specifying weighted graph or binarized graph, subject- or group-level, and the global and/or node summaries used (e.g. clustering coefficient, efficiency, etc.).* |
| Multivariate modeling and predictive analysis | *Specify independent variables, features extraction and dimension reduction, model, training and evaluation metrics.* |

