## [Peer Review File · Nature Structural & Molecular Biology]

Membrane insertion of mitochondrial-encoded proteins regulates ribosome decoding speed

Corresponding Author: Professor Peter Rehling

Version 0:

Decision Letter:

16th Jul 2025

Dear Dr. Rehling,

Thank you for submitting your manuscript "Membrane insertion of mitochondrial-encoded proteins regulates ribosome decoding speed". I apologize for the delay in sharing our decision. I'm writing to let you know that we have decided to send your manuscript for peer review, but several points require your attention before we can proceed with peer review.

I am re-opening the manuscript submission system for you to resubmit your manuscript with all associated files needed for peer review directly, within 2-3 business days if possible. Please follow the link at the bottom of this email to upload the documents listed below. If you have any issues, please reach out to us before completing the submission.

1- We require official wwPDB validation reports for newly described atomic structures, as per journal policy. We also request that authors provide cryo-EM maps, half-maps and models, as well as maps and models obtained from subtomogram averaging if it applies to the work, to help the reviewers in assessing the work. We recommend the use of figshare in our system, which allows for provision of anonymous access links for the referees (<https://www.springernature.com/gp/authors/research-data/figshare-integration>). Alternatively, please upload .zip folders directly with the submission. Please do not upload individual .pdb and .mrc files into the manuscript tracking system.

To ensure the ease of reviewer access to the data, please specify in the Data Availability section where the files can be found (i.e., provide a figshare link or direct the reader to the manuscript files).

2- We want to ensure that the methods and statistics reporting in our papers are of the highest quality. To that end, we ask authors to fill out a Reporting Summary that collects information on experimental design and reagents. If your paper includes ChIP-seq, flow cytometry or MRI data, we ask you take special care to complete those sections of the Reporting Summary as this data will aid greatly in the review of your manuscript. This document can be found by following the link below:

Reporting Summary:

3- In order for us to proceed with peer review, please provide accession numbers and reviewer tokens to access sequencing or proteomics datasets if any unpublished datasets are part of your study. Please add this information to your manuscript file.

4- Lastly, I would like to kindly request that you provide the code used to analyse the data to the reviewers, if newly developed (unpublished) code was used in the work. For the reviewers to evaluate the work adequately, they must be able to test the software/review the code themselves. If you have not yet provided the software, we therefore request that you provide a single compressed zip file containing the software with a readme.txt file or other user manual containing complete instructions for installing and running the software. If appropriate, please provide example data and expected output. Sufficient material should be provided for referees to directly test the performance of the software/algorithm. If the software and materials are small enough to fit in a single compressed zip file under 6MB in size, you may email this file directly to me. If the zip file is between 6 MB and 200 MB, you may upload it to our file transfer site. If necessary, a second zip file up to 200 MB in size can be used to supply the example data. Please let me know if you need to use this option and I'll send you further details. Alternatively, you can upload the code to GitHub and provide us with the link.

Please fill out and return to me the code and software submission checklist that will be made available to editors and reviewers during manuscript assessment. Please note that this form is a dynamic 'smart pdf' and must therefore be

downloaded and completed in Adobe Reader, instead of opening it in a web browser.
<https://www.nature.com/documents/nr-software-policy.pdf>

Please use the link below to submit the files. **Please also remember to move forward all other files associated with this version of the paper.**

Link Redacted

Sincerely,

Katarzyna Ciazynska, PhD
(she/her)
Senior Editor
Nature Structural & Molecular Biology
<https://orcid.org/0000-0002-9899-2428>

Version 1:

Decision Letter:

17th Sep 2025

Dear Dr. Rehling,

Thank you again for submitting your manuscript "Membrane insertion of mitochondrial-encoded proteins regulates ribosome decoding speed" to the journal.

Please accept our sincere apologies for the length of time your manuscript has been under review at our journal. This is due to the unforeseen inability of Reviewer #2 to provide the referee report in a timely manner. As the expertise of the other referees can cover the major aspects of the current study, we have reached our decision based on their comments to avoid further delay. If/when we receive the report from Reviewer #2 within a reasonable timeframe, we will send the comments to you, and in that case we would also expect revisions to address Reviewer #2's comments').

Nevertheless, in light of the reports on hand, we remain interested in your study and would like to see your response to the comments of the referees in the form of a revised manuscript.

You will see that while reviewers appreciate the results, they raise several concerns which will need to be addressed in a revision. Specifically you will see that reviewer #1 states that the generalisability of the findings is speculative at this point. It is our editorial opinion that further analysis in this direction will strengthen the manuscript, and should be explored. We also agree with reviewer 1 that further mutagenesis experiments should be employed to deepen the analysis of stalling/slowing-down. Please provide the clarifications requested by reviewers #3 and #4, and provide additional statistical analysis suggested by reviewer #3.

Please be sure to address/respond to **all** concerns of the referees in full in a point-by-point response and highlight all changes in the revised manuscript text file. We expect to see your revised manuscript within 2-3 months. We are committed to providing a fair and constructive peer-review process; please do not hesitate to contact us if you would like to discuss the reviews or anticipate any delay with the resubmission. If you cannot send it within this time, please contact us to discuss an extension; we will still consider your revision provided that no similar work has been accepted for publication at NSMB or published elsewhere.

REPORTING AND DATA AVAILABILITY: Please provide updated files as follows with your revision. As you already know, we put great emphasis on ensuring that the methods and statistics reported in our papers are correct and accurate.

1. **REPORTING SUMMARY:** Please provide an updated file:

This form is a dynamic 'smart pdf' and must be downloaded and completed in Adobe Reader.

2. REPORTING OF STRUCTURAL DATA: If there are additional or modified structures in the revision, please submit the corresponding PDB validation reports with your other files, and use the figshare integration system to provide access to maps, half-maps, and models.

Manuscripts reporting new structures should contain a table summarizing structural and refinement statistics. Please include these tables for new cryo-EM or -ET (<https://www.nature.com/documents/nr-tables-cryo-em.doc> and modifying this file for ET), NMR (<https://www.nature.com/documents/nr-tables-nmr.doc>) and X-ray crystallography (<https://www.nature.com/documents/nr-tables-xray.doc>). To facilitate assessment of the quality of the structural data, a stereo image of a portion of the electron density map (for crystallography papers) or of the superimposed lowest energy structures (>10; for NMR papers) should be provided with the submitted manuscript. If the reported structure represents a novel overall fold, a stereo image of the entire structure (as a backbone trace) should also be provided. For cryo-EM structures, a representative micrograph showing individual particles should be provided in the figures alongside a processing workflow (please see our Editorial for more information: <https://www.nature.com/articles/s41594-025-01567-9>).

DATA DEPOSITION: We require deposition of coordinates (and, in the case of crystal structures, structure factors) into the Protein Data Bank with the designation of immediate release upon publication (HPUB). Electron microscopy-derived density maps and coordinate data must be deposited in EMDB and released upon publication. Deposition and immediate release of NMR chemical shift assignments are requested. To avoid delays in publication, dataset accession numbers must be supplied with the revised manuscript and appropriate release dates must be indicated at the galley proof stage.

3. REPORTING OF LIGHT MICROSCOPY DATA: For any revision that includes light microscopy data, we ask our authors to please include a completed light microscopy reporting table [https://www.nature.com/documents/Light_microscopy_reporting_table.xlsx] to ensure the methods are described thoroughly. The table will be available to reviewers and

ultimately published, should the manuscript be accepted at the journal. When submitting the revised version of your manuscript, please pay close attention to our [href="https://www.nature.com/nature-portfolio/editorial-policies/image-integrity"](https://www.nature.com/nature-portfolio/editorial-policies/image-integrity)>Digital Image Integrity Guidelines.

4. CODE AND COMPUTATIONAL WORK: If newly developed, unpublished code was used in this work, it must be provided with the revision and access to it must be disclosed in the Code Availability Statement in the manuscript. Please also provide the completed software submission checklist. This form is a dynamic 'smart pdf' and must be downloaded and completed in Adobe Reader, instead of opening it in a web browser: <https://www.nature.com/documents/nr-software-policy.pdf>

If molecular dynamics (MD) simulations were performed, please refer to this Editorial and provide a completed MD simulations checklist with your revision: <https://www.nature.com/articles/s42003-023-04653-0>

Lastly, if it applies, please complete and upload the completed machine learning checklist with your submission for review: <https://www.nature.com/documents/machine-learning-checklist.pdf>

5. SOURCE GEL IMAGES: Unprocessed scans should be provided for all gels and western blots presented in figures and should be clearly labelled with the figure number in the file title. The source gel images should be uncropped, unmodified images, with molecular weight markers. Uncropped, unmodified gel images from existing experimental repeats, in addition to the data shown in the figures, can also be presented in source data. Please provide these files as PDF with your revision.

Please ensure that all control panels for gels and western blots are appropriately described as loading or IP controls. Lastly, please ensure that all images in the paper are checked for duplication of panels and for splicing of gel lanes.

6. SOURCE NUMERICAL DATA: we urge authors to provide, in tabular form in excel files, the data underlying the graphical representations used in figures. This is to further increase transparency in data reporting, as detailed in this editorial (<http://www.nature.com/nsmb/journal/v22/n10/full/nsmb.3110.html>). Please provide one excel file per figure max, with one panel per tab. When submitting files, the title field should indicate which figure the source data pertains to.

7. DATA AVAILABILITY: this journal strongly supports public availability of data. All data used in accepted papers should be available via a public data repository, or alternatively, as Supplementary Information. If data can only be shared on request, please explain why in your Data Availability Statement and in the correspondence with your editor.

For some data types, deposition in a public repository is mandatory as detailed below:

<https://www.nature.com/nature-research/editorial-policies/reporting-standards#availability-of-data>

EXTENDED DATA FIGURES

Link Redacted

Sincerely,

Katarzyna Ciazynska, PhD
(she/her)
Senior Editor
Nature Structural & Molecular Biology
<https://orcid.org/0000-0002-9899-2428>

Referee expertise:

Referee #1: riboseq, ribosomes

Referee #2: cryo-EM, ribosomes

Referee #3: cryo-EM, mitoribosome, mitochondrial biology

Referee #4: mitochondrial protein translocation and QC

Reviewers' Comments:

Reviewer #1 (Remarks to the Author):

This is an excellent and timely study by the Rehling's group that addresses the poorly understood coupling of mitochondrial translation to membrane insertion of nascent chains. By combining mitochondrial ribosome profiling with cryo-EM analysis of COX1 ribosome-nascent chain complexes, the authors convincingly demonstrate that nascent chain topology and amphipathic motifs modulate ribosome elongation speed, resulting in stalling or slow-down events. The structural data

elegantly show how the vestibule of the mitoribosome exit tunnel interacts with nascent helices, in collaboration with the OXA1L/MITRAC machinery, to facilitate co-translational insertion and folding. The work provides conceptual advances in our understanding of how mitochondrial-encoded hydrophobic proteins are synthesized and inserted into the inner membrane, and will be of broad interest to those studying translation, membrane protein biogenesis, and mitochondrial function.

Major suggestions:

1. While COX1 is the logical model, the extent to which the findings generalize to other mitochondrial-encoded proteins remains somewhat speculative. The ribosome profiling data suggest broader principles, but additional validation with another protein (e.g., ND subunit) would strengthen the claim of a universal mechanism. At least, the authors could add some further discussion to the corresponding section.
2. The morpholino-precursor approach is clever, and has been already published by this group. More detail on the efficiency, specificity, and possible off-target effects (particularly whether reduced reads on non-target transcripts are technical artifacts vs. biological regulation) would help the non-experts to better understand the system.
3. I am not a hands-on expert on cryo-EM, but in my opinion the assignment of the low-resolution densities to OXA1L/MITRAC is reasonable. However, given the limited resolution the authors should temper clearly state limitations, especially when describing the “open” versus “closed” states.
4. The data convincingly show stalling/slow-down linked to amphipathic motifs. However, whether this regulation is required for correct folding and insertion, or simply a byproduct, is not fully resolved. Could mutational analysis of these motifs (or vestibule residues) provide functional insight?

Minor Points:

1. A short discussion of how their profiling approach compares quantitatively to other recent mitochondrial profiling protocols (Itoh, Greber, etc.) would help readers gauge improvements.
2. Typographical detail: in the results section “we taligned all mRNAs...” (line 185) should be corrected.

Reviewer #3 (Remarks to the Author):

In the manuscript ‘Membrane insertion of mitochondrial-encoded proteins regulates ribosome decoding speed’ by Schöndorf et al., the authors combine ribosome profiling with structural studies to understand how translation dynamics in human mitochondria are regulated in response to the properties of the synthesized polypeptide. Using selective profiling of MITRAC-bound, COX1 translating mitoribosomes, they find that the protein topology and more specifically the emergence of TMDs from the polypeptide exit tunnel causes slow down of polypeptide synthesis. They use single particle cryo-EM to visualize these stalled translation intermediates and identify a vestibule formed by ribosomal proteins around the polypeptide exit tunnel that senses the structural features of the emerging nascent chain and may control translation speed and the accessibility of the nascent chain to auxiliary factors for protein folding or modification.

The author’s claims are validated by the experimental data and the study is well written and illustrated. Overall, the quality of the manuscript is high and I only have few comments that may need attention prior to publication.

1. Line 185: spelling error: taligned instead of aligned

2. Fig. 4b: I think it is a bit difficult to follow how the authors identify ribosome and C12ORF62-bound COX1 translation intermediate from the images and the text. How do the authors explain the presence of translation products in the puromycin treated C12ORF62 eluate? Is the biogenesis factor involved in other assembly processes? If so, how do they identify in the minus puromycin sample which of the fragments correspond to COX1 and which ones not? By comparison to 4a? If not, how do they explain the copurified S35-labelled fragments? There is also a mistake in the figure legend. It states that the fragments are labelled with red stars while they use green asterisks.

3. The figure legend for Fig. 4c is missing.

4. Fig. 4d: It is difficult to interpret the diagram. Maybe the authors can explain their rationale a bit more clearly. From what I understand, the green lines indicate the position of the P site mRNA codon corresponding to the identified COX1 polypeptide fragment. Yet, the figure legend states that the red arrows show the fragment length and codon position. However, red arrows and green lines do not overlap. It gets more confusing when the authors describe in line 283 that the red triangles are correlated with structural polypeptide features emerging from the exit tunnel. Can the authors please clarify what exactly they show and label?

5. Line 276-277: From the graph, it looks like throughout the transcript high number of reads cannot only be found at the beginning of the black transmembrane segments but more generally at the beginning of segments (no matter if TM or loop) especially in the trace of COX1. Can the authors statistically verify their claim that reads are enriched at the beginning of the TM segments?

6. Figure 5a: The graphs are of poor quality and should be replaced.

7. Figure 5c: Why are 3 curves shown per graph? Are these the biological replicates? Can the authors label it in the figure legend?

8. Line 310-311: How do the authors explain that the stalling at 50 codons downstream is increased for N-OUT mRNAs?

Maybe because the second TMD is then inserted like N-IN?

9. Line 328-330: Can the authors explain why they switched the detergent for the cryo-studies and if they verified that they can isolate the same complexes in both cases?

10. Line 514-517: Although the idea that an open mtRNC conformation may facilitate access of auxiliary factors for protein folding and modification is tempting, the authors do not have experimental evidence for this. I would suggest phrasing this statement therefore rather as a speculation than a fact.

11. Suppl Fig. 2: The graphs are of very poor resolution. Can the authors replace them with better images?

12. Table S1 appears to be missing information regarding B factor used for map sharpening and everything related to the refinement and validation of the structural models. The table needs to be completed prior to publication.

Reviewer #4 (Remarks to the Author):

The mitochondrial genome encodes for 13 proteins of the OXPHOS complexes. The mitochondrial ribosomes associate with the inner membrane to allow their co-translational insertion via OXAL1. OXA1L binds to MITRAC components to couple the import of these proteins with their first assembly steps. The molecular mechanisms of the co-translational protein import into the mitochondrial inner membrane are poorly understood.

Schöndorf and colleagues use an elegant approach to shed light into this fundamental scientific question. By coupling ribosome profiling and cryo-EM structures they found that the presence of membrane domains affects the translation speed. They could show that an amphipathic helix is formed within the vestibule of the ribosome and causes pauses of the translation. The identified mechanism represents a new mode of protein insertion.

The presented data are of high quality and provides new insights into biogenesis of mitochondrial encoded proteins. The findings are exciting for a broad readership of NSMB. I have a few recommendations to improve clarity of the study.

1. The identification of the helix that forms in the ribosome vestibule and its effect on protein translation is exciting. Can the authors discuss whether this molecular mechanism is specific for human mitoribosomes or present in other organelles as well.
2. The structures in Fig. 6 are small and should be enlarged. It is confusing the orientation of the ribosomes is switched between Fig. 6A and Fig. 6C and 6D. I recommend to turn the ribosomes in Fig. 6A to avoid confusion.
3. The work-flow presented in Fig. 1A should be briefly explained in the text.
4. Correct typo in line 185 "taligned".
5. Explain abbreviations like RPKM.

Version 2:

Decision Letter:

Our ref: NSMB-A51375B

30th Jan 2026

Dear Dr. Rehling,

Thank you for submitting your revised manuscript "Membrane insertion of mitochondrial-encoded proteins regulates ribosome decoding speed" (NSMB-A51375B). It has now been seen by the original referees and their comments are below. The reviewers find that the paper has improved in revision, and therefore we'll be happy in principle to publish it in Nature Structural & Molecular Biology, pending minor revisions to satisfy the referees' final requests and to comply with our editorial and formatting guidelines.

We are now performing detailed checks on your paper and will send you a checklist detailing our editorial and formatting requirements in about 2-3 weeks. Please do not upload the final materials and make any revisions until you receive this additional information from us.

Sincerely,

Katarzyna Ciazynska, PhD
(she/her)

Senior Editor
Nature Structural & Molecular Biology
<https://orcid.org/0000-0002-9899-2428>

Reviewer #1 (Remarks to the Author):

The authors have satisfactorily responded to all previous criticism. This is an outstanding piece of work beautifully tailored for NSMB.

Reviewer #2 (Remarks to the Author):

The authors have addressed most of the raised points appropriately.

Reviewer #3 (Remarks to the Author):

The authors have addressed my previous comments.

I have only one minor comments left:

Fig. 3 a,d,g,j show data for ribosomal footprints on all mRNAs in the presence and absence of ND2 morpholino and ribosomal enrichment on the ND2 transcript in the presence of COX1, COX2, and ND2 morpholinos. However, there appears to be no mentioning of the ND2 data in the whole section 'In organello silencing affects ribosome occupancy on target mRNAs', which leaves the panels with no connection to the text. I would suggest that the authors include 1 or 2 sentences to connect the images to the main text in case they want to include the ND2 data. The figure panels referenced in the section appear in parts also not to correlate with what is discussed. That would need to be corrected.

Reviewer #4 (Remarks to the Author):

The authors addressed all my comments. The revised manuscript improved. The reported data are of excellent quality and the findings are highly interesting for a broad readership. I recommend publication of the manuscript in NSMB.

Version 3:

Decision Letter:

31st Mar 2026

Dear Dr. Rehling,

We are now happy to accept your revised paper "Membrane insertion of mitochondrial-encoded proteins regulates ribosome decoding speed" for publication as an Article in Nature Structural & Molecular Biology.

As soon as your article is published, you can generate your shareable link by entering the DOI of your article here: http://authors.springernature.com/share. Corresponding authors will also receive an automated email with the shareable link

Your paper will be published online soon after we receive proof corrections and will appear in print in the next available issue. You can find out your date of online publication by contacting the production team shortly after sending your proof corrections.

An online order form for reprints of your paper is available at https://www.nature.com/reprints/author-reprints.html. Please let your coauthors and your institutions' public affairs office know that they are also welcome to order reprints by this method.

Authors may need to take specific actions to achieve compliance with funder and institutional open access mandates. If your research is supported by a funder that requires immediate open access (e.g. according to Plan S principles or the NIH public access policy) then you should select the gold OA route, and we will direct you to the compliant route where possible. Because authors warrant under our subscription licensing terms that they haven't committed to licensing any version of their article under a licence inconsistent with the terms of our agreement – including the applicable embargo period – publication under the subscription model isn't suitable for authors whose funders require no embargo.

Sincerely,

Katarzyna Ciazynska, PhD
(she/her)
Senior Editor
Nature Structural & Molecular Biology
<https://orcid.org/0000-0002-9899-2428>
